# PERSONALIZE SEGMENT ANYTHING MODEL WITH ONE SHOT

**Renrui Zhang**[1,2], **Zhengkai Jiang**[*3], **Ziyu Guo**[*2], **Shilin Yan**[2], **Junting Pan**[1]
**Hao Dong**[4], **Yu Qiao**[2], **Peng Gao**[2], **Hongsheng Li**[†1,5]

[1]CUHK MMLab    [2]Shanghai Artificial Intelligence Laboratory
[3]Institute of Automation, Chinese Academy of Sciences
[4]CFCS, School of CS, Peking University    [5]CPII of InnoHK
{renruizhang, ziyuguo}@link.cuhk.edu.hk  hsli@ee.cuhk.edu.hk
kaikaijiang.jzk@gmail.com  {gaopeng, qiaoyu}@pjlab.org.cn

## ABSTRACT

Driven by large-data pre-training, Segment Anything Model (SAM) has been demonstrated as a powerful promptable framework, revolutionizing the segmentation field. Despite the generality, customizing SAM for specific visual concepts without man-powered prompting is under-explored, e.g., automatically segmenting your pet dog in numerous images. In this paper, we introduce a training-free **Per**sonalization approach for SAM, termed **PerSAM**. Given only one-shot data, i.e., a single image with a reference mask, we first obtain a positive-negative location prior for the target concept in new images. Then, aided by target visual semantics, we empower SAM for personalized object segmentation via two proposed techniques: target-guided attention and target-semantic prompting. In this way, we can effectively customize the general-purpose SAM for private use without any training. To further alleviate the ambiguity of segmentation scales, we present an efficient one-shot fine-tuning variant, **PerSAM-F**. Freezing the entire SAM, we introduce a scale-aware fine-tuning to aggregate multi-scale masks, which only tunes *2 parameters* within *10 seconds* for improved performance. To demonstrate our efficacy, we construct a new dataset, PerSeg, for the evaluation of personalized object segmentation, and also test our methods on various one-shot image and video segmentation benchmarks. Besides, we propose to leverage PerSAM to improve DreamBooth for personalized text-to-image synthesis. By mitigating the disturbance of training-set backgrounds, our approach showcases better target appearance generation and higher fidelity to the input text prompt. Code is released at https://github.com/ZrrSkywalker/Personalize-SAM.

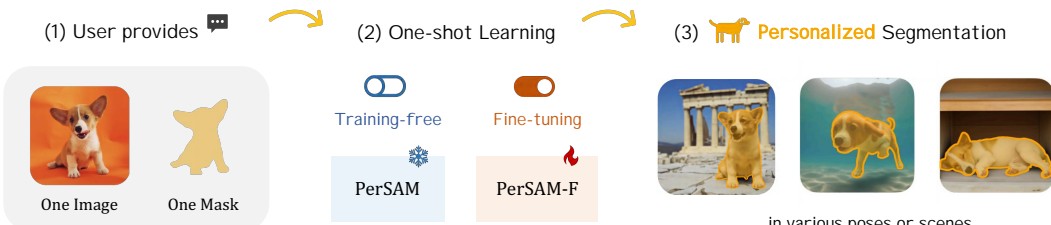

Figure 1: **Personalization of Segment Anything Model.** We customize Segment Anything Model (SAM) (Kirillov et al., 2023) for specific visual concepts, e.g., your pet dog. With only one-shot data, we introduce two efficient solutions: a training-free PerSAM, and a fine-tuning PerSAM-F.

---

* Equal contribution.    † Corresponding author.

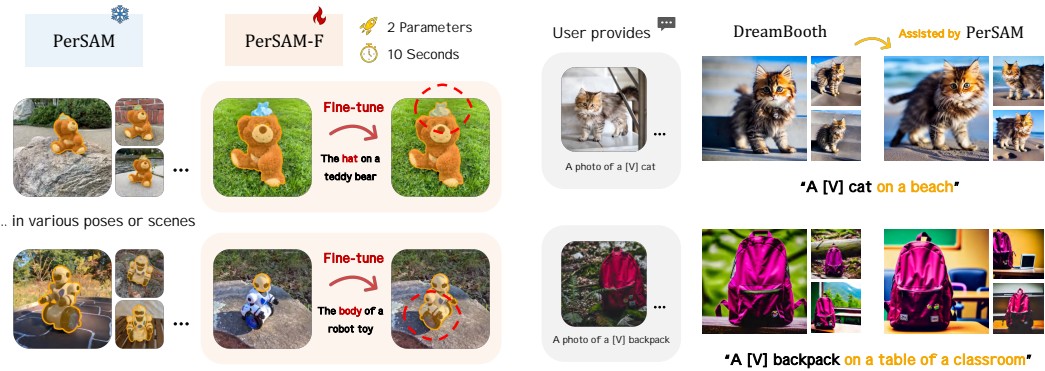

Figure 2: **Personalized Segmentation Examples.** Our PerSAM (Left) can segment personal objects in any context with favorable performance, and PerSAM-F (right) further alleviates the ambiguity issue by scale-aware fine-tuning.

Figure 3: **Improving DreamBooth (Ruiz et al., 2022) with PerSAM.** By mitigating the disturbance of backgrounds during training, our approach can help to achieve higher-quality personalized text-to-image generation.

# 1 INTRODUCTION

Foundations models in vision (Li et al., 2022; Zou et al., 2023; Wang et al., 2022), language (Brown et al., 2020; Touvron et al., 2023; Radford et al., 2019), and multi-modality (Radford et al., 2021; Jia et al., 2021; Li et al., 2023) have gained unprecedented prevalence, attributed to the availability of large-scale datasets and computational resources. They demonstrate extraordinary generalization capacity in zero-shot scenarios, and display versatile interactivity incorporating human feedback. Inspired by this, Segment Anything (Kirillov et al., 2023) develops a delicate data engine for collecting 11M image-mask data, and subsequently trains a segmentation foundation model, known as SAM. It defines a novel promptable segmentation framework, i.e., taking as input a handcrafted prompt and returning the expected mask, which allows for segmenting any objects in visual contexts.

However, SAM inherently loses the capability to segment specific visual concepts. Imagine intending to crop your lovely pet dog in a thick photo album, or find the missing clock from a picture of your bedroom. Utilizing the vanilla SAM would be highly labor-intensive and time-consuming. For each image, you must precisely find the target object within complicated contexts, and then activate SAM with a proper prompt for segmentation. Considering this, we ask: *Can we personalize SAM to automatically segment user-designated visual concepts in a simple and efficient manner?*

To this end, we introduce **PerSAM**, a training-free personalization approach for Segment Anything Model. As shown in Figure 1, our method efficiently customizes SAM using only one-shot data, i.e., a user-provided reference image and a rough mask of the personal concept. Specifically, we first obtain a location confidence map for the target object in the test image by feature similarities, which considers the appearance of every foreground pixel. According to confidence scores, two points are selected as the positive-negative location prior, which are finally encoded as prompt tokens and fed into SAM's decoder for segmentation. Within the decoder, we propose to inject visual semantics of the target object to unleash SAM's personalized segmentation power with two techniques:

- **Target-guided Attention.** We guide every token-to-image cross-attention layer in SAM's decoder by the location confidence map. This explicitly compels the prompt tokens to mainly concentrate on foreground target regions for intensive feature aggregation.

- **Target-semantic Prompting.** To explicitly provide SAM with high-level target semantics, we fuse the original prompt tokens with the embedding of the target object, which provides the low-level positional prompt with additional visual cues for personalized segmentation.

With the aforementioned designs, along with a cascaded post-refinement, PerSAM exhibits favorable personalized segmentation performance for unique subjects in a variety of poses or scenes. Notably, our approach can cope well with scenarios that require segmenting one object among multiple similar ones, simultaneously segmenting several identical objects in the same image, or tracking different objects along a video. Nevertheless, as shown in Figure 2, there might be occasional failure cases,

where the object comprises visually distinct subparts or hierarchical structures to be segmented, e.g., the hat on top of a teddy bear, or the head of a robot toy. Such ambiguity casts a challenge for PerSAM in determining the appropriate scale of mask as output, since both the local part and the global shape can be regarded as valid masks by SAM.

To alleviate this issue, we further propose a fine-tuning variant of our approach, **PerSAM-F**. We freeze the entire SAM to preserve its versatile pre-trained knowledge, and only fine-tune *2 parameters* within *10 seconds* on a single A100 GPU. In detail, we enable SAM to produce several potential segmentation results of different mask scales. To adaptively select the best scale for varying objects, we employ a learnable relative weight for each mask scale, and conduct a weighted summation as the final output. By such efficient scale-aware training, PerSAM-F avoids over-fitting on the one-shot data and exhibits better segmentation accuracy shown in Figure 2 (Right).

Moreover, we observe that our approach can also assist DreamBooth (Ruiz et al., 2022) to better fine-tune diffusion models for personalized text-to-image generation, as shown in Figure 3. Given a few images containing a specific visual concept, e.g., your pet cat or backpack, DreamBooth learns to convert these images into an identifier [V] in the word embedding space, which, however, can simultaneously include the background information, e.g., stairs or the forest. This would override the newly prompted backgrounds, and disturb the target appearance generation. Therefore, we propose to leverage PerSAM to segment the target object within training images, and only supervise DreamBooth by the foreground area, enabling text-to-image synthesis with higher quality.

We summarize the contributions of our paper as follows:

- **Personalized Object Segmentation.** We first investigate how to customize a general-purpose segmentation model (SAM) into personalized scenarios with minimal expense. To this end, we introduce two efficient and effective methods, along with a new segmentation dataset, PerSeg, for the evaluation of personalized object segmentation.

- **PerSAM and PerSAM-F.** In PerSAM, we propose three training-free techniques to guide SAM by the high-level semantics of target objects. In PerSAM-F, we design a scale-aware fine-tuning with 2 parameters in 10 seconds to well alleviate the mask ambiguity issue.

- Our approach achieves competitive results on various tasks, including the PerSeg benchmark, one-shot part and semantic segmentation, and video object segmentation. In addition, PerSAM can enhance DreamBooth for better personalized text-to-image synthesis.

## 2 METHOD

In Section 2.1, we first briefly revisit Segment Anything Model (SAM) (Kirillov et al., 2023), and introduce the task definition for personalized object segmentation. Then, we illustrate the methodology of our PerSAM and PerSAM-F in Section 2.2 and 2.3, respectively. Finally, we utilize our approach to assist DreamBooth (Ruiz et al., 2022) for better text-to-image generation in Section 2.5, and specifically discuss some application scenarios in Section 2.4.

### 2.1 PERSONALIZED OBJECT SEGMENTATION

**A Revisit of Segment Anything.** SAM consists of three components, a prompt encoder, an image encoder, and a lightweight mask decoder, respectively denoted as $\mathrm{Enc}_P$, $\mathrm{Enc}_I$, and $\mathrm{Dec}_M$. As a promptable framework, SAM takes as input an image $I$, and a set of prompts $P$, which can be a point, a box, or a coarse mask. Specifically, SAM first utilizes $\mathrm{Enc}_I$ to obtain the input image feature, and adopts $\mathrm{Enc}_P$ to encode the human-given prompts of a length $k$ into prompt tokens as

$$F_I = \mathrm{Enc}_I(I), \quad T_P = \mathrm{Enc}_P(P), \tag{1}$$

where $F_I \in \mathbb{R}^{h \times w \times c}$ and $T_P \in \mathbb{R}^{k \times c}$, with $h, w$ denoting the resolution of the image feature map and $c$ denoting the feature dimension. After that, the encoded image and prompts are fed into the decoder $\mathrm{Dec}_M$ for attention-based feature interaction. SAM constructs the input tokens of the decoder by concatenating several learnable mask tokens $T_M$ as prefixes to the prompt tokens $T_P$. These mask tokens are responsible for generating the mask output, formulated as

$$M = \mathrm{Dec}_M \Big( F_I, \ \mathrm{Concat}(T_M, T_P) \Big), \tag{2}$$

where $M$ denotes the final segmentation mask predicted by SAM.

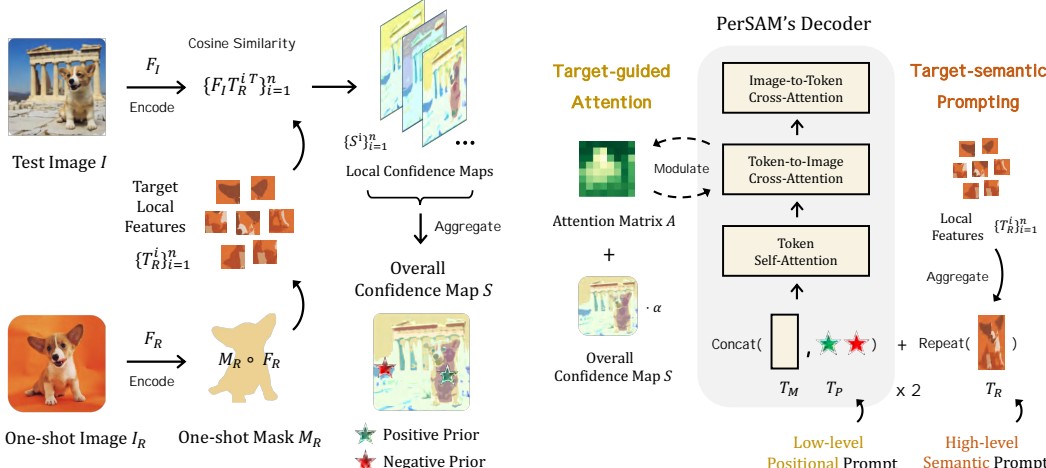

Figure 4: **Positive-negative Location Prior.** We calculate a location confidence map for the target object in new test image by the appearance of all local parts. Then, we select the location prior as the point prompt for PerSAM.

Figure 5: **Target-guided Attention (Left) & Target-semantic Prompting (Right).** To inject SAM with target semantics, we explicitly guide the cross-attention layers, and propose additional prompting with high-level cues.

**Task Definition.** Although SAM is generalized enough for any object by prompting, it lacks the ability to automatically segment specific subject instances. Considering this, we define a new task for personalized object segmentation. The user provides only a single reference image, and a mask indicating the target visual concept. The given mask can either be an accurate segmentation, or a rough sketch drawn on-the-fly. Our goal is to customize SAM to segment the designated object within new images or videos, without additional human prompting. For evaluation, we annotate a new dataset for personalized segmentation, named PerSeg. The raw images are collected from the works for subject-driven diffusion models (Gal et al., 2022; Ruiz et al., 2022; Kumari et al., 2022), containing various categories of visual concepts in different poses or scenes. In this paper, we propose two efficient solutions for this task, which we specifically illustrate as follows.

## 2.2 TRAINING-FREE PERSAM

**Location Confidence Map.** Conditioned on the user-provided image $I_R$ and mask $M_R$, PerSAM first obtains a confidence map that indicates the location of the target object in the new test image $I$. As shown in Figure 4, we apply an image encoder to extract the visual features of both $I_R$ and $I$. The encoder can be SAM's frozen backbone or other pre-trained vision models, for which we adopt SAM's image encoder $\mathrm{Enc}_I$ by default. We formulate the process as

$$F_I = \mathrm{Enc}_I(I), \quad F_R = \mathrm{Enc}_I(I_R), \tag{3}$$

where $F_I, F_R \in \mathbb{R}^{h \times w \times c}$. Then, we utilize the reference mask $M_R \in \mathbb{R}^{h \times w \times 1}$ to crop the features of foreground pixels within the visual concept from $F_R$, resulting in a set of $n$ local features as

$$\{T_R^i\}_{i=1}^n = M_R \circ F_R, \tag{4}$$

where $T_R^i \in \mathbb{R}^{1 \times c}$ and $\circ$ denotes spatial-wise multiplication. After this, we calculate $n$ confidence maps for each foreground pixel $i$ by the cosine similarity between $T_R^i$ and test image feature $F_I$ as

$$\{S^i\}_{i=1}^n = \{F_I T_R^{i\,T}\}_{i=1}^n, \quad \text{where } S^i \in \mathbb{R}^{h \times w}. \tag{5}$$

Note that $F_I$ and $T_R^i$ have been pixel-wisely L2-normalized. Each $S^i$ represents the distribution probability for a different local part of object in the test image, such as the head, the body, or the paws of a dog. On top of this, we adopt an average pooling to aggregate all $n$ local maps to obtain the overall confidence map of the target object as

$$S = \frac{1}{n} \sum_{i=1}^n S^i \in \mathbb{R}^{h \times w}. \tag{6}$$

By incorporating the confidences of every foreground pixel, $S$ can take the visual appearance of different object parts into consideration, and acquire a relatively comprehensive location estimation.

**Positive-negative Location Prior.** To provide PerSAM with a location prior on the test image, we select two points with the highest and lowest confidence values in $S$, denoted as $P_h$ and $P_l$, respectively. The former represents the most likely center position of the target object, while the latter inversely indicates the background. Then, they are regarded as the positive and negative point prompts, and fed into the prompt encoder as

$$T_P = \text{Enc}_P(P_h, P_l) \in \mathbb{R}^{2 \times c}, \tag{7}$$

which denote the prompt tokens for SAM's decoder. In this way, SAM would tend to segment the contiguous region surrounding the positive point, while discarding the negative one's on the image.

**Target-guided Attention.** Although the positive-negative point prompt has been obtained, we further propose a more explicit semantic guidance to the cross-attention operation in SAM's decoder, which concentrates the feature aggregation within foreground target regions. As shown in Figure 5, the overall confidence map $S$ in Equation 6 can clearly indicate the rough region of the target visual concept in the test image (hotter colors indicate higher scores). Based on such a property, we utilize $S$ to guide the attention map in every token-to-image cross-attention layer of the decoder. Specifically, we denote every attention map after the softmax function as $A \in \mathbb{R}^{h \times w}$, and then modulate its attention distribution by

$$A^g = \text{softmax}\left(A + \alpha \cdot \text{softmax}(S)\right), \tag{8}$$

where $\alpha$ denotes a balancing factor. With the attention bias, the mask and prompt tokens are compelled to capture more visual semantics associated with the target subject, other than the unimportant background area. This contributes to more effective feature aggregation in attention mechanisms, and enhances the final segmentation accuracy of PerSAM in a training-free manner.

**Target-semantic Prompting.** The vanilla SAM only receives prompts with low-level positional information, such as the coordinate of a point or a box. To provide SAM's decoder with more high-level cues, we propose to utilize the visual feature of the target concept as an additional high-level semantic prompting. We first obtain the global embedding $T_R$ of the object in the reference image by both average pooling between different local features as

$$T_R = \frac{1}{n} \sum_{i=1}^{n} T_R^i \in \mathbb{R}^{1 \times c}. \tag{9}$$

Then, we element-wisely add $T_R$ to all the input tokens of the test image in Equation 2, before feeding them into the decoder block, which is shown in Figure 5 as

$$T^g = \text{Repeat}(T_R) + \text{Concat}(T_M, T_P), \tag{10}$$

where $T^g$ denotes the input token guided by target semantics for the decoder $\text{Dec}_M$, and the $\text{Repeat}$ operation duplicates the target visual embedding. Aided by the simple token incorporation, PerSAM is not only prompted by low-level location points, but also high-level target visual cues.

**Cascaded Post-refinement.** Via the above techniques, we obtain an initial segmentation mask on the test image from SAM's decoder, which however, might include rough edges and isolated background noises. For further refinement, we iteratively feed the mask back into the decoder $\text{Dec}_M$ for a two-step post-processing. In the first step, we prompt the decoder by the currently predicted mask along with the previous positive-negative point prompt. For the second step, we acquire the bounding box enclosing the mask from the first step, and prompt the decoder additionally with this box for more accurate object localization. As we only iterate the lightweight decoder without the large-scale image encoder, the post-processing is efficient and only costs an extra 2% latency.

## 2.3 FINE-TUNING OF PERSAM-F

**Ambiguity of Segmentation Scales.** The training-free PerSAM can tackle most cases with satisfactory segmentation accuracy. However, some target objects contain hierarchical structures, which leads

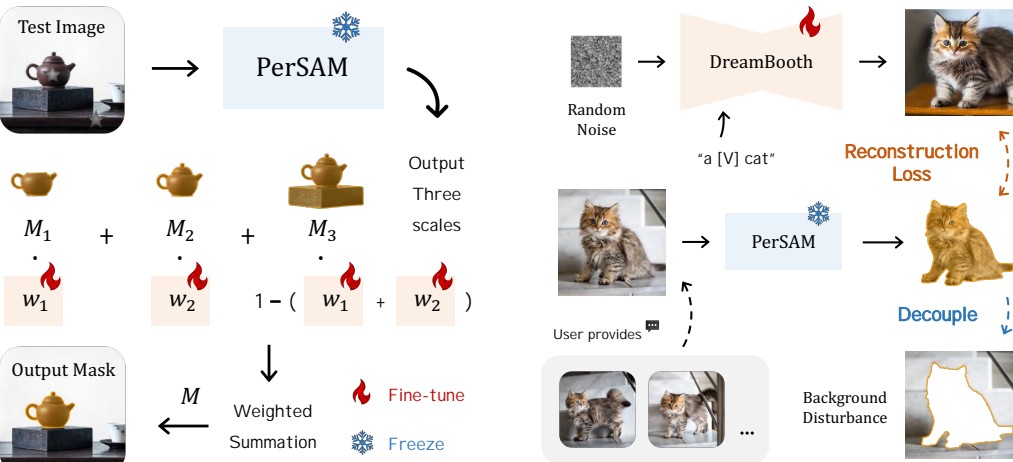

Figure 6: **The Scale-aware Fine-tuning in PerSAM-F.** To alleviate the scale ambiguity, PerSAM-F adopts two learnable weights for adaptively aggregating three-scale masks.

Figure 7: **PerSAM-assisted DreamBooth.** We utilize PerSAM to decouple the target objects from the background for improving the generation of DreamBooth.

to the ambiguity of mask scales. As shown in Figure 6, the teapot on top of a platform is comprised of two parts: a lid and a body. If the positive point prompt (denoted by a green pentagram) is located at the body, while the negative prompt (denoted by a red pentagram) does not exclude the platform in a similar color, PerSAM would be misled for segmentation. Such an issue is also discussed in SAM, where it proposes an alternative to simultaneously generate multiple masks of three scales, i.e., the whole, part, and subpart of an object. Then, the user must manually select one mask out of three, which is effective but consumes extra manpower. In contrast, our personalized task aims to customize SAM for automatic object segmentation without the need for human prompting. This motivates us to further develop a scale-aware version of PerSAM by parameter-efficient fine-tuning.

**Scale-aware Fine-tuning.** For adaptive segmentation with the appropriate scale, we introduce a fine-tuning variant, PerSAM-F. Unlike the training-free model only producing one mask, PerSAM-F first follows PerSAM to obtain the location prior, and refers to SAM's original solution to output three-scale masks, denoted as $M_1$, $M_2$, and $M_3$, respectively. On top of this, we adopt two learnable mask weights, $w_1, w_2$, and calculate the final mask output by a weighted summation as

$$M = w_1 \cdot M_1 + w_2 \cdot M_2 + (1 - w_1 - w_2) \cdot M_3, \quad (11)$$

where $w_1, w_2$ are both initialized as $1/3$. To learn the optimal weights, we conduct one-shot fine-tuning on the reference image, and regard the given mask as the ground truth. Note that, we freeze the entire SAM model to preserve its pre-trained knowledge, and only fine-tune the **2 parameters** of $w_1, w_2$ within **10 seconds** on a single A100 GPU. In this way, our PerSAM-F efficiently learns the scale-aware semantics of objects, and adaptively outputs the best segmentation scale for different concepts, improving the generalization capacity of PerSAM.

## 2.4 PerSAM-assisted DreamBooth

For personalized text-to-image synthesis, DreamBooth (Ruiz et al., 2022) fine-tunes a pre-trained diffusion model by the given 3~5 photos of a specific object, i.e., a pet cat. It learns to generate the cat referred to by a text prompt, "a [V] cat", and calculates the loss over the entire reconstructed images. This This would inject the redundant background information in the training images into the identifier [V]. Therefore, as shown in Figure 7, we introduce our strategy to alleviate the disturbance of backgrounds in DreamBooth. Given an object mask for any of the few-shot images, we leverage our PerSAM to segment all the foreground targets, and discard the gradient back-propagation for pixels belonging to the background area. Then, the Stable Diffusion is only fine-tuned to memorize the visual appearances of the target object. With no supervision imposed on the background, our PerSAM-assisted DreamBooth can not only synthesize the target object with better visual correspondence, but also increase the diversity of the new backgrounds guided by the input text prompt.

Table 1: **Personalized Object Segmentation on the PerSeg Dataset**. We compare the overall mIoU, bIoU, and learnable parameters for different methods (Bar et al., 2022; Wang et al., 2022; 2023; Zou et al., 2023), along with the mIoU for 10 objects in PerSeg. '*' denotes works concurrent to ours.

| Method | mIoU | bIoU | Param. | Can | Barn | Clock | Cat | Back-pack | Teddy Bear | Duck Toy | Thin Bird | Red Cartoon | Robot Toy |
|---|---|---|---|---|---|---|---|---|---|---|---|---|---|
| Painter | 56.4 | 42.0 | 354M | 19.1 | 3.2 | 42.9 | 94.1 | 88.1 | 93.0 | 33.3 | 20.9 | 98.2 | 65.0 |
| VP | 65.9 | 25.5 | 383M | 61.2 | 58.6 | 59.2 | 76.6 | 66.7 | 79.8 | 89.9 | 67.4 | 81.0 | 72.4 |
| SEEM* | 87.1 | 55.7 | 341M | 65.4 | 82.5 | 72.4 | 91.1 | 94.1 | 95.2 | 98.0 | 71.3 | 97.0 | 95.8 |
| SegGPT* | 94.3 | 76.5 | 354M | 96.6 | 63.8 | 92.6 | 94.1 | 94.4 | 93.7 | 97.2 | 92.6 | 97.3 | 96.2 |
| **PerSAM** | 89.3 | 71.7 | 0 | 96.2 | 38.9 | 96.2 | 90.70 | 95.39 | 94.6 | 97.3 | 93.7 | 97.0 | 60.6 |
| **PerSAM-F** | 95.3 | 77.9 | 2 | 96.7 | 97.5 | 96.1 | 92.3 | 95.5 | 95.2 | 97.3 | 94.0 | 97.1 | 96.7 |

Table 2: **Video Object Segmentation** on DAVIS 2017 val (Pont-Tuset et al., 2017). We utilize gray color to denote the methods involving in-domain training.

| Method | $\mathcal{J}\&\mathcal{F}$ | $\mathcal{J}$ | $\mathcal{F}$ |
|---|---|---|---|
| AGSS | 67.4 | 64.9 | 69.9 |
| AFB-URR | 74.6 | 73.0 | 76.1 |
| Painter | 34.6 | 28.5 | 40.8 |
| SEEM | 58.9 | 55.0 | 62.8 |
| SegGPT | 75.6 | 72.5 | 78.6 |
| **PerSAM** | 66.9 | 63.4 | 70.4 |
| **PerSAM-F** | 76.1 | 73.2 | 78.9 |

Table 3: **One-shot Semantic and Part Segmentation** on FSS-1000 (Li et al., 2020), LVIS-92$^i$ (Gupta et al., 2019), PASCAL-Part (Morabia et al., 2020), and PACO-Part (Ramanathan et al., 2023). We report the mIoU scores and utilize gray color to denote the methods involving in-domain training.

| Method | One-shot Semantic Seg. | | One-shot Part Seg. | |
|---|---|---|---|---|
| | FSS-1000 | LVIS-92$^i$ | PASCAL-Part | PACO-Part |
| HSNet | 86.5 | 17.4 | 32.4 | 22.6 |
| VAT | 90.3 | 18.5 | 33.6 | 23.5 |
| Painter | 61.7 | 10.5 | 30.4 | 14.1 |
| SegGPT | 85.6 | 18.6 | - | - |
| **PerSAM** | 81.6 | 15.6 | 32.5 | 22.5 |
| **PerSAM-F** | 86.3 | 18.4 | 32.9 | 22.7 |

## 2.5 DISCUSSION

**Can PerSAM segment multiple *Different* objects?**  *Yes.* If the user indicates three different objects in the reference image, we can also personalize SAM with these three objects, as visualized in Figure 9 for video object segmentation. For a new test image, we first adopt the image encoder to extract its visual feature only once. On top of this, we independently calculate the location confidence maps for the three objects, and then prompt the decoder three times to respectively segment them. As the decoder (50ms) is more lightweight than the encoder (2s), this brings a marginal time cost.

**Comparison with Concurrent Works.**  Some concurrent works, such as SegGPT (Wang et al., 2023) and SEEM (Zou et al., 2023), can also segment objects given one-shot data, i.e., a reference image and a mask. However, our goal is to customize an off-the-shelf generalist (SAM) into a specialist for private use with good efficiency (training-free or 10-second fine-tuning). In contrast, they target generalization capabilities requiring extensive training data optimizing large-scale parameters. Moreover, due to the efficient tuning characters, PerSAM can be regarded as a general extensible framework, acceptable for any image encoder and promptable segmentation models. Therefore, these excellent works and our approach are trying to solve different problems in segmentation fields.

## 3 EXPERIMENT

We first evaluate our approach for personalized segmentation on PerSeg in Section 3.1, along with various existing one-shot segmentation benchmarks in Section 3.2. Then, we illustrate the effectiveness of our PerSAM-assisted DreamBooth in Section 3.3. Finally, we conduct several ablation studies to investigate our designs on PerSeg in Section 3.4.

## 3.1 PERSONALIZED EVALUATION

**PerSeg Dataset.**  To test the personalization capacity, we construct a new segmentation dataset, termed PerSeg. The raw images are collected from the training data of subject-driven diffusion works (Ruiz et al., 2022; Gal et al., 2022; Kumari et al., 2022). PerSeg contains 40 objects of various

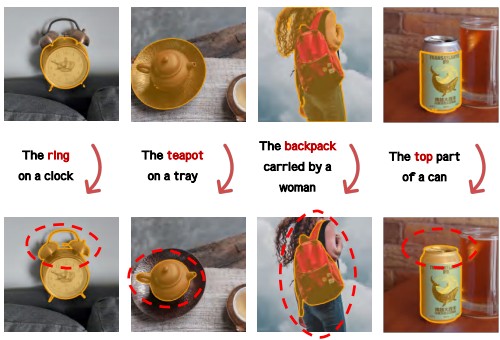 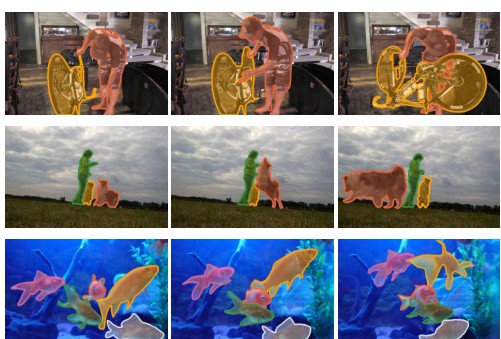

Figure 8: **Visualization of PerSAM-F's Improvement.** Our scale-aware fine-tuning can well alleviate the scale ambiguity of PerSAM.

Figure 9: **Visualization of Video Object Segmentation.** Our approach performs well for segmenting multiple objects in a video.

categories in total, including daily necessities, animals, and buildings. In different poses or scenes, each object is associated with 5∼7 images and masks, where we fix one image-mask pair as the user-provided one-shot data. The mIoU and bIoU (Cheng et al., 2021a) are adopted for evaluation.

**Performance.** In Table 1, we observe the fine-tuned PerSAM-F achieves the best results, which effectively enhances PerSAM by +2.7% and +5.9% overall mIoU and bIoU. We show more visualization of PerSAM-F's improvement in Figure 8. Visual Prompting (VP) (Bar et al., 2022), Painter (Wang et al., 2022), SEEM (Zou et al., 2023), and SegGPT (Wang et al., 2023) are in-context learners that can also segment objects according to the given one-shot prompt data. As shown, the training-free PerSAM can already achieve better performance than Painter, VP, and SEEM with different margins. By the efficient 2-parameter fine-tuning, our PerSAM-F further surpasses the powerful SegGPT by +2.4% and +4.1% overall mIoU and bIoU. As analyzed in Section 2.5, different from their motivations, our method is specially designed for personalized object segmentation, and exhibits much more efficiency in both time and computational resources.

## 3.2 EXISTING SEGMENTATION BENCHMARKS

**Video Object Segmentation.** Given the first-frame image and object masks, our PerSAM and PerSAM-F achieve competitive object segmentation and tracking performance on the validation set of DAVIS 2017 (Pont-Tuset et al., 2017) As shown in Table 2, compared to methods without video training, the training-free PerSAM largely surpasses Painter by +32.3% $\mathcal{J}\&\mathcal{F}$ score, and our PerSAM-F can achieve +0.5% better performance than SegGPT. Notably, our one-shot fine-tuning approach can outperform methods (Lin et al., 2019; Liang et al., 2020) fully trained by extensive video data. The results fully illustrate our strong generalization ability for temporal video data and complex scenarios, which contain multiple similar or occluded objects, as visualized in Figure 9.

**One-shot Semantic and Part Segmentation.** In Table 3, we evaluate our approach for one-shot image segmentation respectively on four datasets, FSS-1000 (Li et al., 2020), LVIS-92$^i$ (Gupta et al., 2019), PASCAL-Part (Morabia et al., 2020), and PACO-Part (Ramanathan et al., 2023), where we follow (Liu et al., 2023b) for data pre-processing and evaluation. As shown, our PerSAM-F attains consistently better results than Painter, and performs comparably to SegGPT. For models (Min et al., 2021; Hong et al., 2022) with in-domain training, our approach can achieve higher scores than HSNet. The experiments well demonstrate that, our proposed approach is not limited to object-level segmentation, but also works for category-wise and part-wise personalization of SAM.

## 3.3 PERSAM-ASSISTED DREAMBOOTH

We follow all the hyperparameters in DreamBooth (Ruiz et al., 2022) to fine-tune a pre-trained Stable Diffusion (Rombach et al., 2022) for personalized image synthesis. In addition to Figure 3, we visualize more examples of PerSAM-assisted DreamBooth in Figure 10. For the dog lying on a grey sofa, the "jungle" and "snow" by DreamBooth are still the sofa with green and white decorations. Assisted by PerSAM-F, the newly-generated background is totally decoupled with the sofa and well

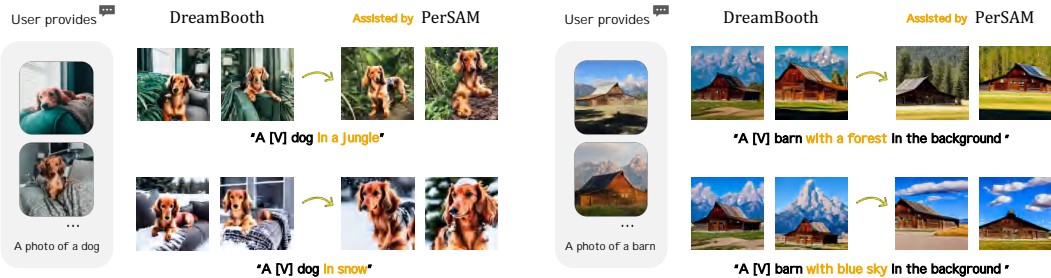

Figure 10: **Visualization of PerSAM-guided DreamBooth.** The improved DreamBooth (Ruiz et al., 2022) can better preserve the diversity for synthesizing various contexts in new images.

Table 4: **Ablation of Main Components** in our proposed method.

| Variant | mIoU | Gain |
|---|---|---|
| Positive Prior | 69.1 | - |
| + Negative Prior | 72.5 | +3.4 |
| + Post-refinement | 83.9 | +11.4 |
| + Guided Attention | 85.8 | +1.9 |
| + Semantic Prompt | 89.3 | +3.5 |
| + Scale Tuning | 95.3 | +6.0 |

Table 5: **Ablation of Different Fine-tuning Methods**.

| Method | Param. | mIoU |
|---|---|---|
| PerSAM | 0 | 89.32 |
| Prompt Tuning | 12K | 76.5 |
| Adapter | 196K | 78.3 |
| LoRA | 293K | 90.0 |
| 3 Mask Weights | 3 | 92.9 |
| PerSAM-F | 2 | 95.3 |

Table 6: **Ablation of using Box-image as Reference**.

| Method | Mask | Box |
|---|---|---|
| Painter | 56.4 | 42.0 |
| VP | 65.9 | 38.1 |
| SEEM | 87.1 | 64.9 |
| SegGPT | 94.3 | 36.0 |
| PerSAM | 89.3 | 88.1 |
| PerSAM-F | 95.3 | 94.9 |

corresponds to the textual prompt. For the barn in front of the mountains, our approach also alleviates the background disturbance to correctly generate the "forest" and "blue sky".

## 3.4 ABLATION STUDY

**Main Components.** In Table 4, we investigate our different components by starting from a baseline that only adopts the positive location prior. Then, we add the negative point prompt and cascaded post-refinement, enhancing +3.6% and +11.4% mIoU, respectively. On top of that, we introduce the high-level target semantics into SAM's decoder for attention guidance and semantic prompting. The resulting +1.9% and +3.5% improvements fully indicate their significance. Finally, via the efficient scale-aware fine-tuning, PerSAM-F boosts the score by +6.0%, demonstrating superior accuracy.

**Different Fine-tuning Methods.** In Table 5, we experiment with other parameter-efficient fine-tuning (PEFT) methods for PerSAM-F, i.e., prompt tuning (Liu et al., 2021), Adapter (Houlsby et al., 2019), and LoRA (Hu et al., 2021). We freeze the entire SAM, and only tune the PEFT modules injected into every transformer block in PerSAM's decoder. As shown, the prompt tuning and Adapter would over-fit the one-shot data and severely degrade the accuracy. Instead, our scale-aware fine-tuning can best improve the performance of PerSAM, while tuning the least learnable parameters.

**Using Box-image as Reference.** Requiring an accurate mask as one-shot data might be too strict for some users. In Table 6, we relax the input restrictions to a bounding box designating the expected object. For our method, we can regard the box as a prompt and utilize off-the-shelf SAM to generate the one-shot mask. Therefore, the box reference only leads to a marginal performance drop in PerSAM and PerSAM-F, but severely influences other methods.

## 4 CONCLUSION

In this paper, we propose to personalize Segment Anything Model (SAM) for specific visual concepts with only one-shot data. Firstly, we introduce PerSAM, which injects high-level target semantics into SAM with training-free techniques. On top of this, we present a scale-aware fine-tuning variant, PerSAM-F. With only 2 learnable parameters, PerSAM-F effectively alleviates the ambiguity of mask scales and achieves leading performance on various benchmarks. Besides, we also verify the efficacy of our approach to assist DreamBooth in fine-tuning better text-to-image diffusion models. We hope our work may expand the applicability of SAM to a wider range of scenarios.

## 5 ACKNOWLEDGEMENT

This work is partially supported by the National Key R&D Program of China (NO.2022ZD0161100), the National Natural Science Foundation of China (No.62206272), the Centre for Perceptual and Interactive Intelligence (CPII) Ltd under the Innovation and Technology Commission (ITC)'s InnoHK, and General Research Fund of Hong Kong RGC Project 14204021. Hongsheng Li is a PI of CPII under the InnoHK.

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

## A    OVERVIEW

## B    RELATED WORK

**Foundation Models.**    With powerful generalization capacity, pre-trained foundation models can be adapted for various downstream scenarios and attain promising performance. In natural language processing, BERT (Devlin et al., 2018; Lu et al., 2019), GPT series (Brown et al., 2020; OpenAI, 2023; Radford & Narasimhan, 2018; Radford et al., 2019), and LLaMA (Touvron et al., 2023) have demonstrated remarkable in-context learning abilities, and can be extended to multi-modal scenarios (Zhang et al., 2023c; Gao et al., 2024; Lin et al., 2023; Han et al., 2023; Guo et al., 2023; Zhang et al., 2024). Similarly, CLIP (Radford et al., 2021) and ALIGN (Jia et al., 2021), which conduct contrastive learning on image-text pairs, exhibit exceptional accuracy in zero-shot visual recognition. Painter (Wang et al., 2022) introduces a vision model that unifies network architectures and in-context prompts to accomplish diverse vision tasks, without downstream fine-tuning. CaFo (Zhang et al., 2023d) cascades different foundation models and collaborates their pre-trained knowledge for robust low-data image classification. SAM (Kirillov et al., 2023) presents a foundation model for image segmentation, which is pre-trained by 1 billion masks and conducts prompt-based segmentation. There are some concurrent works extending SAM for high-quality segmentation (Ke et al., 2023), faster inference speed (Zhao et al., 2023; Zhang et al., 2023a), all-purpose matching (Liu et al., 2023b), 3D reconstruction (Cen et al., 2023), object tracking (Yang et al., 2023), medical (Ma & Wang, 2023; Huang et al., 2023) image processing. From another perspective, we propose to personalize the segmentation foundation model, i.e., SAM, for specific visual concepts, which adapts a generalist into a specialist with only one shot. Our method can also assist the personalization of text-to-image foundation models, i.e., Stable Diffusion (Rombach et al., 2022) and Imagen (Saharia et al., 2022), which improves the generation quality by segmenting the foreground target objects from the background disturbance.

**Large Models in Segmentation.**    As a fundamental task in computer vision, segmentation (Long et al., 2015; Jiang et al., 2022; Zhao et al., 2017; Xu et al., 2021; Jiang et al., 2023; Lin et al., 2022) requires a pixel-level comprehension of a image. Various segmentation-related tasks have been explored, such as semantic segmentation, classifying each pixel into a predefined set of classes (Badrinarayanan et al., 2017; Chen et al., 2017; Zheng et al., 2021; Cheng et al., 2022; Xie et al., 2021; Song et al., 2020b); instance segmentation, focusing on the identification of individual object instances (He et al., 2017; Wang et al., 2020; Tian et al., 2020a); panoptic segmentation, assigning both class labels and instance identification (Kirillov et al., 2019; Li et al., 2019); and interactive segmentation, involving human intervention for refinement (Hao et al., 2021; Chen et al., 2021). Recently, inspired by language foundation models (Zhang et al., 2023c; Brown et al., 2020), several concurrent works have proposed large-scale vision models for image segmentation. They are pre-trained by extensive mask data and exhibit strong generalization capabilities on numerous image distributions. Segment Anything Model (SAM) (Kirillov et al., 2023) utilizes a data engine with model-in-the-loop annotation to learn a promptable segmentation framework, which generalizes to downstream scenarios in a zero-shot manner. Painter (Wang et al., 2022) and SegGPT (Wang et al., 2023) introduce a robust in-context learning paradigm and can segment any images by a given image-mask prompt. SEEM (Zou et al., 2023) further presents a general segmentation model prompted by multi-modal references, e.g., language and audio, incorporating versatile semantic knowledge. In this study, we introduce a new task termed personalized object segmentation, and annotate a new dataset PerSeg for evaluation. Instead of developing large segmentation models, our goal is to personalize them to segment user-provided objects in any poses or scenes. We propose two approaches, PerSAM and PerSAM-F, which efficiently customize SAM for personalized segmentation.

**Parameter-efficient Fine-tuning.**    Directly tuning the entire foundation models on downstream tasks can be computationally expensive and memory-intensive, posing challenges for resource-

Table 7: **Personalized Object Segmentation on the PerSeg Dataset**. We report the mIoU scores of 30 objects in addition to the 10 objects in Table 1. '*' denotes works concurrent to ours.

| Method | Dog | Dog2 | Dog3 | Dog4 | Dog5 | Dog6 | Tortoise Plushy | Round Bird | Colorful Sneaker | Colorful Teapot |
|---|---|---|---|---|---|---|---|---|---|---|
| Painter (Wang et al., 2022) | 80.41 | 73.77 | 46.98 | 22.39 | 82.03 | 76.16 | 55.31 | 39.83 | 0.00 | 13.69 |
| VP (Bar et al., 2022) | 6.80 | 12.26 | 23.84 | 20.61 | 21.44 | 32.05 | 24.42 | 34.09 | 30.32 | 34.89 |
| SEEM* (Zou et al., 2023) | 71.04 | 35.35 | 67.02 | 81.87 | 75.02 | 72.99 | 78.75 | 38.74 | 20.08 | 44.44 |
| SegGPT* (Wang et al., 2023) | 73.82 | 65.07 | 61.11 | 81.66 | 82.94 | 76.44 | 77.85 | 82.89 | 72,24 | 80.44 |
| **PerSAM** | 96.79 | 95.66 | 88.85 | 95.22 | 97.10 | 94.66 | 93.06 | 96.79 | 94.48 | 96.27 |
| **PerSAM-F** | 96.81 | 95.79 | 88.67 | 95.18 | 97.22 | 94.85 | 97.09 | 96.85 | 95.13 | 84.41 |

| Method | Dog7 | Dog8 | Candle | Fancy Boot | Sloth Plushie | Poop Emoji | Rc Car | Shiny Sneaker | Wolf Plushie | Wooden Pot |
|---|---|---|---|---|---|---|---|---|---|---|
| Painter (Wang et al., 2022) | 40.97 | 57.15 | 24.36 | 49.06 | 45.78 | 23.42 | 23.69 | 0.00 | 38.97 | 57.61 |
| VP (Bar et al., 2022) | 17.67 | 12.24 | 12.71 | 39.13 | 29.31 | 37.55 | 29.98 | 30.88 | 28.86 | 34.30 |
| SEEM* (Zou et al., 2023) | 63.77 | 70.34 | 26.99 | 34.90 | 81.46 | 45.55 | 34.94 | 82.30 | 76.27 | 74.81 |
| SegGPT* (Wang et al., 2023) | 66.20 | 82.21 | 81.60 | 76.06 | 80.54 | 81.32 | 79.26 | 85.26 | 72.48 | 78.00 |
| **PerSAM** | 93.69 | 95.34 | 74.16 | 95.87 | 96.37 | 96.01 | 39.30 | 97.00 | 94.34 | 97.42 |
| **PerSAM-F** | 93.77 | 95.61 | 96.75 | 95.96 | 96.64 | 96.43 | 96.12 | 96.87 | 94.32 | 97.43 |

| Method | Table | Teapot | Chair | Elephant | Duck Toy | Monster Toy | Dog Pack | Bear Plushie | Berry Bowl | Cat Statue |
|---|---|---|---|---|---|---|---|---|---|---|
| Painter (Wang et al., 2022) | 16.92 | 7.00 | 50.09 | 40.80 | 29.24 | 34.80 | 40.73 | 81.30 | 45.98 | 19.96 |
| VP (Bar et al., 2022) | 16.00 | 10.00 | 27.20 | 22.01 | 52.14 | 30.92 | 22.80 | 23.95 | 11.32 | 27.54 |
| SEEM* (Zou et al., 2023) | 30.15 | 12.30 | 66.15 | 46.64 | 89.92 | 41.49 | 66.83 | 61.27 | 38.29 | 24.27 |
| SegGPT* (Wang et al., 2023) | 81.95 | 89.89 | 78.97 | 80.38 | 84.48 | 83.33 | 77.53 | 75.54 | 73.00 | 76.54 |
| **PerSAM** | 94.68 | 40.02 | 92.22 | 96.05 | 97.31 | 93.75 | 95.85 | 89.28 | 91.81 | 95.42 |
| **PerSAM-F** | 94.66 | 96.93 | 92.14 | 96.07 | 97.31 | 94.21 | 95.76 | 95.32 | 91.27 | 95.46 |

constrained applications. To address this issue, recent works have focused on developing parameter-efficient methods (Sung et al., 2022; He et al., 2022; Rebuffi et al., 2017; Qin & Eisner, 2021) to freeze the weights of foundation models and append small-scale modules for fine-tuning. Prompt tuning (Lester et al., 2021; Zhou et al., 2022; Jia et al., 2022; Liu et al., 2021) suggests using learnable soft prompts alongside frozen models to perform specific downstream tasks, achieving more competitive performance with scale and robust domain transfer compared to full model tuning. Low-Rank Adaption (LoRA) (Hu et al., 2021; Cuenca & Paul, 2023; Zhang et al., 2023b; Hedegaard et al., 2022) injects trainable rank decomposition matrices concurrently to each pre-trained weight, which significantly reduces the number of learnable parameters required for downstream tasks. Adapters (Houlsby et al., 2019; Pfeiffer et al., 2020; Lin et al., 2020; Chen et al., 2022) are designed to be inserted between layers of the original transformer, introducing lightweight MLPs for feature transformation. Different from existing works, we adopt a more efficient adaption method delicately designed for SAM, i.e., the scale-aware fine-tuning of PerSAM-F with only 2 parameters and 10 seconds. This effectively avoids the over-fitting issue on one-shot data, and alleviates the ambiguity of segmentation scale with superior performance.

## C EXPERIMENTAL DETAILS AND VISUALIZATION

### C.1 PERSONALIZED EVALUATION

**Implementation Details.** We adopt a pre-trained SAM (Kirillov et al., 2023) with a ViT-H (Dosovitskiy et al., 2020) backbone as the foundation model, and utilize SAM's encoder to calculate the location confidence map. For PerSAM, we apply the target-guided attention and target-semantic prompting to all three blocks in the decoder. The balance factor $\alpha$ in Equation 8 is set as 1. For PerSAM-F, we conduct one-shot training for 1,000 epochs with a batch size 1, supervised by the dice loss (Milletari et al., 2016) and focal loss (Lin et al., 2017). We set the initial learning rate as $10^{-3}$, and adopt an AdamW (Loshchilov & Hutter, 2017) optimizer with a cosine scheduler.

**Complete Results on the PerSeg Dataset.** In Table 7, we report the mIoU scores of the other 30 objects in the PerSeg dataset, except for the 10 objects in Table (1) of the main paper. As compared, our PerSAM without any training can achieve superior segmentation results to Painter (Wang et al.,

2022), Visual Prompting (VP) (Bar et al., 2022), and SEEM (Zou et al., 2023) on most objects. Note that, we here compare the results of SEEM with the Focal-L (Yang et al., 2022) vision backbone, its best-performing variant. Aided by the 2-parameter fine-tuning, PerSAM-F further performs comparably with SegGPT (Wang et al., 2023), a powerful in-context segmentation framework. Therefore, our approach exhibits a high performance-efficiency trade-off by efficiently customizing the off-the-shelf SAM (Kirillov et al., 2023) for personalized object segmentation.

**Visualization.** In Figure 11, we visualize the location confidence maps, segmentation results of PerSAM with positive-negative location prior, and the bounding boxes from the cascaded post-refinement. As shown, the confidence map (hotter colors indicate higher scores) can clearly indicate the rough region of the target object in the image, which contributes to precise foreground (green pentagram) and background (red pentagram) point prompts selection. The bounding boxes in green also well enclose the targets and prompt SAM's decoder for accurate post-refinement.

## C.2 EXISTING SEGMENTATION BENCHMARKS

**Implementation Details.** For experiments in existing segmentation datasets, we utilize DI-NOv2 (Oquab et al., 2023) as the image encoder to calculate the location confidence map, which produces a more accurate location prior. Note that, the generality and extensibility of our approach enable us to apply any vision backbones for location confidence map calculation. For video object segmentation, different from PerSeg, where one image contains only one object, DAVIS 2017 dataset (Pont-Tuset et al., 2017) requires to personalize SAM to track and segment multiple different objects across the video frames. In PerSAM, we regard the top-2 highest-confidence points as the positive location prior, and additionally utilize the bounding boxes from the last frame to prompt the decoder. This provides more sufficient temporal cues for object tracking and segmentation. In PerSAM-F, we conduct one-shot fine-tuning on the first frame for 800 epochs with a learning rate $4^{-4}$. As discussed in Section 2.5, for $N$ objects, we only need to run SAM's large-scale encoder (2s) once to encode the visual feature of the new frame, while running the lightweight decoder for $N$ times to segment different objects, which takes marginal $50N$ms. For one-shot semantic segmentation, we evaluate our method on FSS-1000 (Li et al., 2020) following HSNet (Min et al., 2021) and LVIS-$92^i$ (Gupta et al., 2019) pre-processed by (Liu et al., 2023b). The benchmark contains objects in a wide range of semantic categories within various backgrounds. For one-shot part segmentation, we utilize the part-level benchmarks of PASCAL VOC (Morabia et al., 2020) and PACO (Ramanathan et al., 2023) built by (Liu et al., 2023b), requiring to segment partial objects with challenging scenarios.

**Visualization.** In Figure 12, we visualize more results of PerSAM-F for multi-object tracking and segmentation in consecutive frames of the DAVIS 2017 dataset. We utilize different colors to denote different objects, along with the additional prompts for SAM's decoder, including a bounding box from the last frame and its center point. Aided by our techniques and the last-frame temporal cues, PerSAM-F exhibits favorable video segmentation performance and tracking consistency, even for objects occluded by others or objects of the same category with similar appearances. In Figure 13, we also visualize the results of PerSAM-F for one-shot semantic and part segmentation on four datasets. The satisfactory performance illustrates that our approach is not limited to object-level personalization, but also part- and category-wise segmentation with good generalization capability.

## C.3 PERSAM-ASSISTED DREAMBOOTH

**Implementation Details.** We follow most model hyperparameters and training configurations in DreamBooth (Ruiz et al., 2022), including a $10^{-6}$ learning rate and a batch size 1. We generate a 200-image regularization dataset by the pre-trained Stable Diffusion (Rombach et al., 2022) using the textual prompt: "photo of a [CLASS]". We fine-tune DreamBooth and our approach both for 1,000 iterations on a single A100 GPU, and adopt 't@y' as the word identifier [V] for the personal visual concepts. We utilize DDIM (Song et al., 2020a) sampling with 100 steps and a 10-scale classifier-free guidance for generation.

**Quantitative Evaluation.** Besides visualization, we also evaluate the PerSAM-assisted Dream-Booth by three quantitative metrics in Table 19. We leverage CLIP (Radford et al., 2021) to calculate

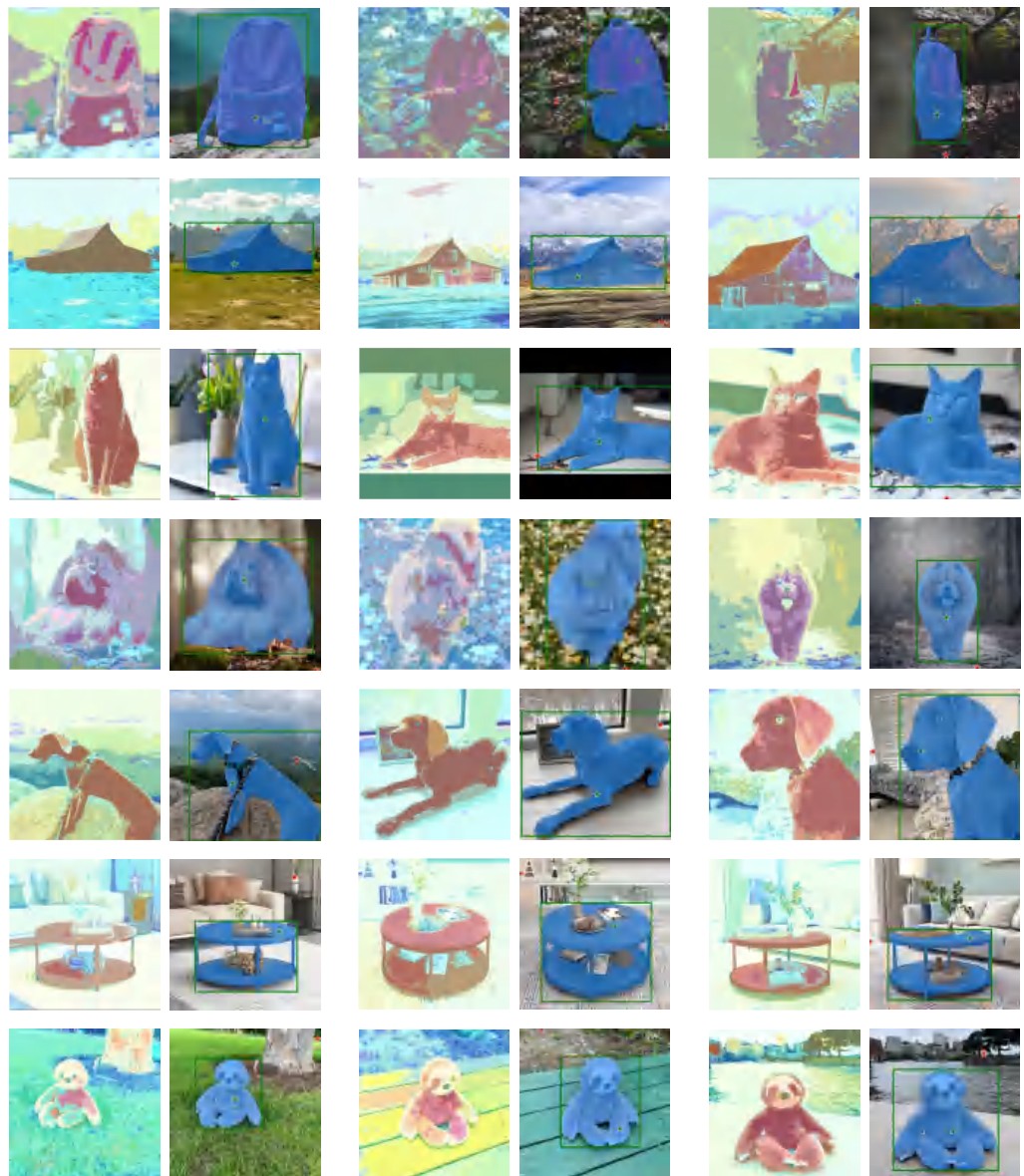

Figure 11: **Visualization of Location Confidence Maps and PerSAM's Segmentation Results.** We represent the positive (foreground) and negative (background) location prior by green and red pentagrams. The green bounding boxes denote the box prompts in the cascaded post-refinement.

the feature similarity of generated images with textual prompts ('Text-Align') and reference images ('Image-Align') (Kumari et al., 2022), along with KID (Bińkowski et al., 2018) (the smaller, the better). 'Text-Align' (Gal et al., 2022) and 'Image-Align' (Hessel et al., 2021) indicate the semantic correspondence of the synthesized images with the textual prompt and few-shot reference images, respectively. KID (Bińkowski et al., 2018) measures how much the fine-tuned models over-fit the specific visual concepts in few-shot images, for which we utilize Stable Diffusion to generate 500 images as the validation set. These quantitative results demonstrate our effectiveness in generating better visual correspondence with the target objects and input prompts.

**Visualization.** In Figure 14, we visualize more examples that demonstrate our effectiveness to enhance DreamBooth for higher-fidelity personalized synthesis. We utilize PerSAM-F to decouple the table and plushy tortoise from their backgrounds in the few-shot images, i.e., the couch and

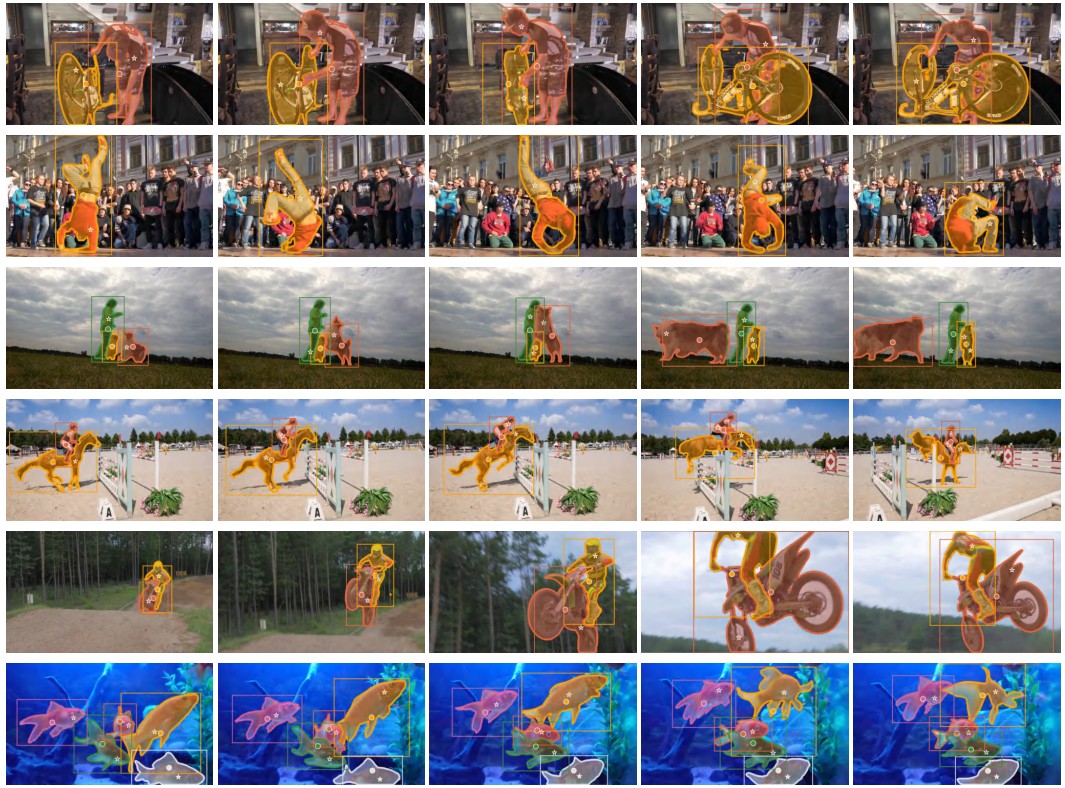

Figure 12: **Visualization of PerSAM-F for Video Object Segmentation** on the DAVIS 2017 (Pont-Tuset et al., 2017) dataset. We represent different objects in different colors, and visualize the input prompts for SAM's decoder: a positive location prior (pentagram), an enclosing bounding box from the last frame, and its center point (dot).

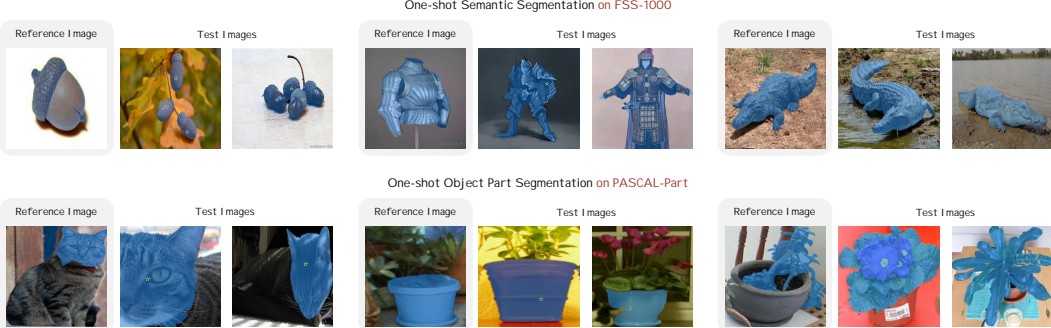

Figure 13: **Visualization of PerSAM-F for One-shot Semantic and Part Segmentation** on FSS-1000 (Li et al., 2020) and PASCAL-Part (Morabia et al., 2020) datasets. Our approach exhibits superior generalization capabilities for diverse segmentation scenarios.

carpet. In this way, the PerSAM-assisted DreamBooth generates new backgrounds corresponding to the textual prompts of "in the garden", "and an orange sofa", "on the grass", and "swimming in a pool". In addition, our approach can boost the appearance generation of target objects with high text-image correspondence, while the vanilla DreamBooth might be interfered by textual prompts, e.g., the orange on the table and the blue on the turtle shell. The experiments fully verify our efficacy for better personalizing text-to-image models.

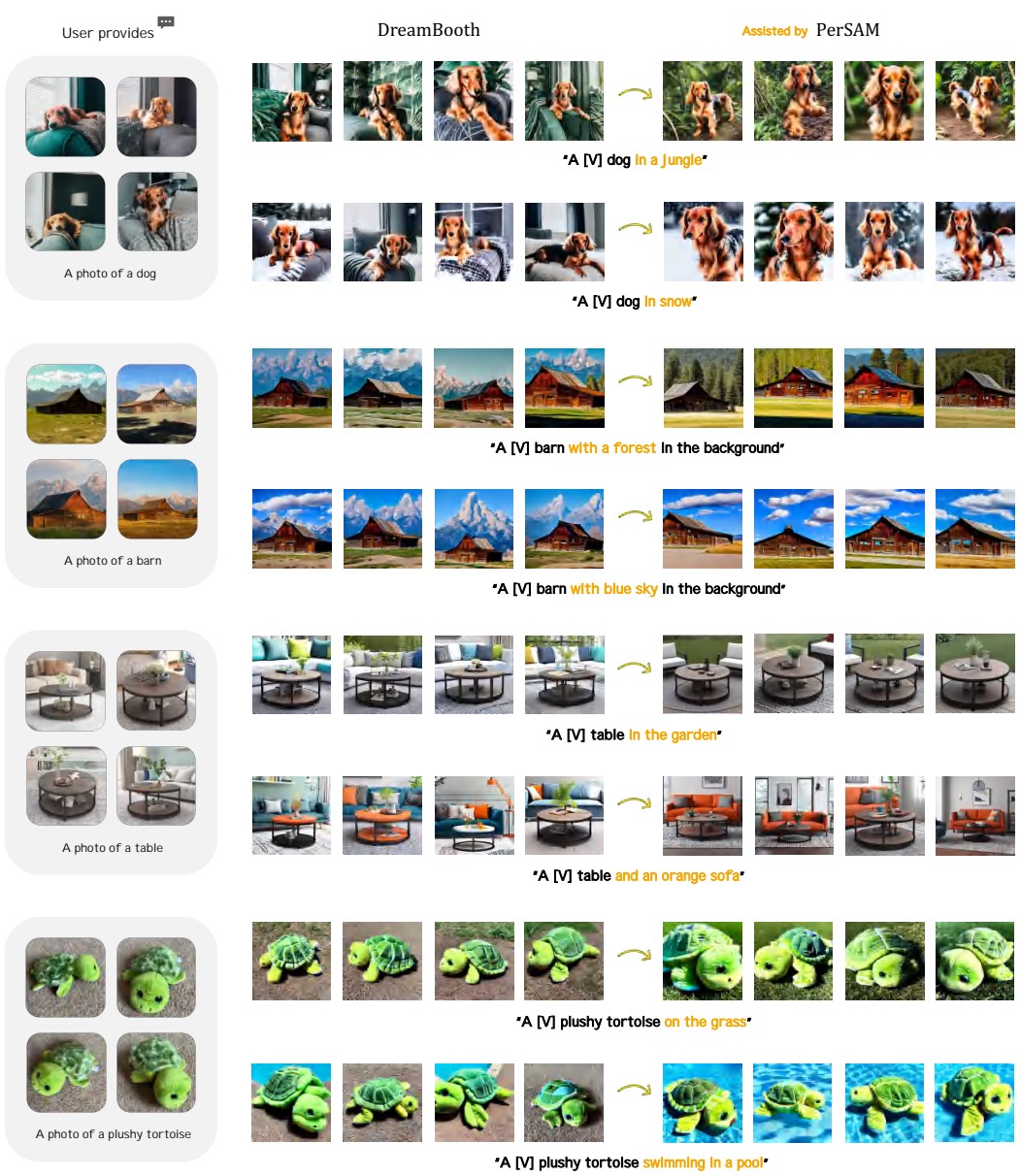

Figure 14: **Visualization of PerSAM-assisted DreamBooth.** Our approach can alleviate the background disturbance, and boost DreamBooth (Ruiz et al., 2022) for better personalized synthesis.

## D ADDITIONAL EXPERIMENTS AND ANALYSIS

### D.1 EVALUATION ON ADDITIONAL BENCHMARKS

**COCO-20[i] (Nguyen & Todorovic, 2019).** Constructed from MSCOCO (Lin et al., 2014), COCO-20[i] divides the diverse 80 classes evenly into 4 folds for one-shot semantic segmentation. We directly test our method on the validation set without specific in-domain training. As shown in Table 8, our PerSAM(-F) achieves favorable segmentation performance over a wide range of object categories, comparable to previous in-domain methods, i.e., FPTrans (Zhang et al., 2022), SCCAN (Xu et al., 2023), and HDMNet (Peng et al., 2023).

**Tokyo Multi-Spectral-4[i] (Bao et al., 2021).** Sampled from Tokyo Multi-Spectral (Ha et al., 2017), Tokyo Multi-Spectral-4[i] contains 16 classes within outdoor city scenes, similar to CityScapes (Cordts

Table 8: **One-shot segmentation on COCO-$20^i$ (Nguyen & Todorovic, 2019).**

| Method | In-domain Train | mIoU |
|---|:---:|:---:|
| FPTrans | ✓ | 47.0 |
| SCCAN | ✓ | 48.2 |
| HDMNet | ✓ | 50.0 |
| PerSAM | - | 47.9 |
| PerSAM-F | - | 50.6 |

Table 9: **One-shot segmentation on Tokyo Multi-Spectral-$4^i$ (Bao et al., 2021).**

| Method | In-domain Train | mIoU |
|---|:---:|:---:|
| PFENet | ✓ | 14.0 |
| PGNet | ✓ | 17.5 |
| V-TFSS | ✓ | 26.1 |
| PerSAM | - | 18.4 |
| PerSAM-F | - | 25.6 |

Table 10: **Comparison with two text-guided models: OVSeg (Liang et al., 2023) and Grounded-SAM (gro, 2023).**

| Method | Prompt | PerSeg | COCO-$20^i$ |
|---|:---:|:---:|:---:|
| OVSeg | Category Name | 76.5 | 37.8 |
| Grounded-SAM | Category Name | 93.2 | 51.3 |
| PerSAM | One-shot Data | 89.3 | 47.9 |
| PerSAM-F | One-shot Data | 95.3 | 50.6 |

Table 11: **Running Efficiency compared to SAM (Kirillov et al., 2023).**

| Method | FPS↑ | Memory (MB)↓ |
|---|:---:|:---:|
| SAM | 2.16 | 5731 |
| PerSAM | 2.08 | 5788 |
| PerSAM-F | 1.98 | 5832 |

Table 12: **Comparison with SAM-PT (Rajič et al., 2023) on DAVIS 2017 (Pont-Tuset et al., 2017).**

| Method | Propagation | J&F |
|---|:---:|:---:|
| SAM-PT | Point Tracking | 76.6 |
| PerSAM | Feature Matching | 66.9 |
| PerSAM | +Point Tracking | 68.2 |
| PerSAM-F | Feature Matching | 76.1 |
| PerSAM-F | +Point Tracking | 77.2 |

et al., 2016). Different from existing methods, we only take as input the RGB images without the paired thermal data, and do not conduct in-domain training. As shown in Table 9, our approach still exhibits good generalization capacity in street scenarios, compared to the specialist models: PFENet (Tian et al., 2020b), PGNet (Zhang et al., 2019), and V-TFSS (Bao et al., 2021).

### D.2 COMPARISON TO ADDITIONAL METHODS

**Text-guided Segmenters.** Recently, open-world segmentation models guided by text prompts have driven increasing attention. To compare our approach with them, we select two popular methods: OVSeg (Liang et al., 2023) and Grounded-SAM (gro, 2023). OVSeg leverages MaskFormer (Cheng et al., 2021b) to first generate class-agnostic mask proposals, and then adopts a fine-tuned CLIP for zero-shot classification. Grounded-SAM utilizes a powerful text-guided detector, Grounding DINO (Liu et al., 2023a), to generate object bounding boxes, and then utilize them to prompt SAM for segmentation. Instead of giving a one-shot reference, we directly prompt them by the category name of the target object for text-guided segmentation, e.g., "cat", "dog", or "chair". As shown in Table 10, our PerSAM-F consistently achieves competitive results on two different datasets: PerSeg and COCO-$20^i$. This indicates that, utilizing PerSAM with a class-agnostic one-shot reference is on par with recognizing the category and then segmenting it with text-guided methods.

**SAM-PT (Rajič et al., 2023).** Although both our PerSAM(-F) and the concurrent SAM-PT are developed based on SAM, our approach can be generalized to most one-shot segmentation tasks (personalized/video/semantic/part segmentation), while SAM-PT specifically aims at video object segmentation. One key difference between our approach and SAM-PT is how to locate and associate

Table 13: **Few-shot segmentation on the PerSeg dataset.**

| Method | Shot | mIoU | bIoU |
|--------|------|------|------|
| SegGPT | 1-shot | 94.3 | 76.5 |
| SegGPT | 3-shot | 96.7 | 78.4 |
| PerSAM | 1-shot | 89.3 | 71.7 |
| PerSAM | 3-shot | 90.2 | 73.6 |
| PerSAM-F | 1-shot | 95.3 | 77.9 |
| PerSAM-F | 3-shot | 97.4 | 79.1 |

Table 14: **Few-shot segmentation on FSS-1000 (Li et al., 2020) benchmark.**

| Method | Shot | mIoU |
|--------|------|------|
| SegGPT | 1-shot | 85.6 |
| SegGPT | 5-shot | 89.3 |
| PerSAM | 1-shot | 81.6 |
| PerSAM | 5-shot | 82.3 |
| PerSAM-F | 1-shot | 86.3 |
| PerSAM-F | 5-shot | 89.8 |

Table 15: **Different pre-trained encoders for obtaining the positive-negative location prior.**

| Method | Encoder | DAVIS 2017 | FSS-1000 | LVIS-92$^i$ | PASCAL-Part | PACO-Part |
|--------|---------|-----------|----------|------------|-------------|-----------|
| Painter | - | 34.6 | 61.7 | 10.5 | 30.4 | 14.1 |
| SegGPT | - | 75.6 | 85.6 | 18.6 | - | - |
| PerSAM | SAM | 62.8 | 74.9 | 12.9 | 31.3 | 21.2 |
| PerSAM | DINOv2 | 66.9 | 81.6 | 15.6 | 32.5 | 22.5 |
| PerSAM-F | SAM | 73.4 | 79.4 | 16.2 | 32.0 | 21.3 |
| PerSAM-F | DINOv2 | 76.1 | 86.3 | 18.4 | 32.9 | 22.7 |

objects from the previous to the current frame, i.e., propagating the location prompt for SAM across frames. In detail, our PerSAM(-F) simply calculates a location confidence map by feature matching, while SAM-PT relies on an external point tracking network, PIPS (Harley et al., 2022). As shown in Table 12, on DAVIS 2017 dataset (Pont-Tuset et al., 2017), SAM-PT performs slightly better than the original PerSAM-F. However, inspired by SAM-PT, we can also incorporate its point tracking strategy (the PIPS tracker) with PerSAM(-F) to propagate the positive-negative point prompt, which effectively enhances the segmentation performance. This demonstrates the flexible extensibility of our approach for applying more advanced trackers in a plug-and-play way.

### D.3 Few-shot Segmentation by PerSAM

Our approach is not limited to one-shot segmentation, and can accept few-shot references for improved results. As an example, given 3-shot references, we independently calculate 3 location confidence maps for the test image, and adopt a pixel-wise max pooling to obtain the overall location estimation. For PerSAM-F, we regard all 3-shot data as the training set to conduct the scale-aware fine-tuning.

We respectively conduct experiments for 3-shot segmentation on PerSeg dataset and 5-shot segmentation on FSS-1000 dataset (Li et al., 2020). The results are respectively shown in Tables 13 and 14. By providing more visual semantics in few-shot data, both our training-free PerSAM and the fine-tuned PerSAM-F can be further enhanced.

### D.4 Ablation Study

**Different Pre-trained Encoders.** For video object segmentation in Table 2 and other one-shot segmentation in Table 3, we adopt the DINOv2 (Oquab et al., 2023) encoder to obtain the positive-negative location prior by default. In Table 15, we show the results by using SAM's original image encoder. As DINOv2 is particularly pre-trained by large-scale contrastive data, it produces more discriminative image features than SAM's encoder. This contributes to a more precise positive-negative location prior for better segmentation results, especially on the challenging FSS-1000 dataset (Li et al., 2020). Despite this, with SAM's original encoder, our PerSAM-F and the training-free PerSAM still obtain better segmentation accuracy than Painter (Wang et al., 2022) or SEEM (Zou et al., 2023) on different datasets, demonstrating the effectiveness of our approach.

Table 16: **Different Image Encoders** of SAM for PerSAM and PerSAM-F.

| Method | Encoder | mIoU | bIoU |
|--------|---------|------|------|
| PerSAM | ViT-B | 63.98 | 49.30 |
|        | ViT-L | 86.61 | 69.86 |
|        | ViT-H | **89.32** | **71.67** |
| PerSAM-F | ViT-B | 87.24 | 69.36 |
|          | ViT-L | 92.24 | 75.36 |
|          | ViT-H | **95.33** | **77.92** |

Table 17: **Robustness to Mask Reference**. We resize the reference mask by 'erode' and 'dilate' functions in OpenCV (Bradski, 2000).

| Method | Shrink↑↑ | Shrink↑ | Enlarge↑ | Enlarge↑↑ |
|--------|----------|---------|----------|-----------|
| SegGPT | 80.39 | 81.79 | 83.22 | 76.43 |
| **PerSAM** | 78.48 | 81.10 | **89.32** | **88.92** |
| **PerSAM-F** | **85.16** | **88.28** | 83.19 | 81.19 |

**Image Encoders of SAM.** By default, we adopt a pre-trained ViT-H (Dosovitskiy et al., 2020) in SAM as the image encoder for PerSAM and PerSAM-F. In Table 16, we investigate the performance of other vision backbones for our models, i.e., ViT-B and ViT-L. As shown, stronger image encoders lead to higher segmentation mIoU and bIoU scores. When using ViT-B as the encoder, the accuracy of training-free PerSAM is largely harmed, due to weaker feature encoding ability, while the one-shot training of PerSAM-F can effectively mitigate the gap by +23.26% mIoU and +20.06% bIoU scores, which demonstrates the significance of our fine-tuning on top of a weak training-free baseline.

**Robustness to the Quality of Mask Reference.** For more robust interactivity with humans, we explore how our approach performs if the given mask is of low quality. In Table 17, we respectively shrink and enlarge the area of the reference mask, and compare the results on PerSeg dataset. When the mask is smaller than the target object (shrink), PerSAM-F, aided by one-shot fine-tuning, exhibits the best robustness. In this case, the target embedding cannot incorporate complete visual appearances from the reference image, which largely harms the training-free techniques in PerSAM. When the mask becomes larger (enlarge), the oversize mask would mislead the scale-aware training of PerSAM-F. In contrast, despite some background noises, the target embedding can include all the visual semantics of objects, which, thereby, brings little influence to PerSAM.

# E DISCUSSION

## E.1 WHAT'S THE ADDITIONAL RUNNING SPEED/MEMORY COMPARED TO SAM?

We test the additional running consumption of PerSAM and PerSAM-F on a single NVIDIA A100 GPU with batch size 1. As shown in Table 11, our PerSAM and PerSAM-F bring marginal latency and GPU memory consumption over SAM, indicating superior running efficiency.

## E.2 HOW TO DIFFERENTIATE SIMILAR OBJECTS IN VIDEO OBJECT SEGMENTATION?

For video object segmentation, our approach tries to accurately locate the target object among similar ones by the following three aspects.

**Discriminative Features from the Encoder.** Due to large-scale pre-training, the SAM's image encoder, or the more powerful DINOv2, can already produce discriminative visual features for different similar objects, which is fundamental to the calculation of location confidence map.

**Comprehensive Location Confidence Map.** We calculate a set of confidence maps for all foreground pixels within the target object, such as the head, the body, or the paws of a dog, and then aggregate them to obtain an overall location estimation. This strategy can comprehensively consider the slight differences in any local parts between similar objects.

**Temporal Cues between Adjacent Frames.** To better leverage the temporal consistency along the video, we prompt SAM's decoder additionally with the object bounding box from the last frame. As different objects have different trajectories, such temporal constraints can better differentiate similar objects by spatial locations.

One-shot Segmentation in Outdoor Street Scenes

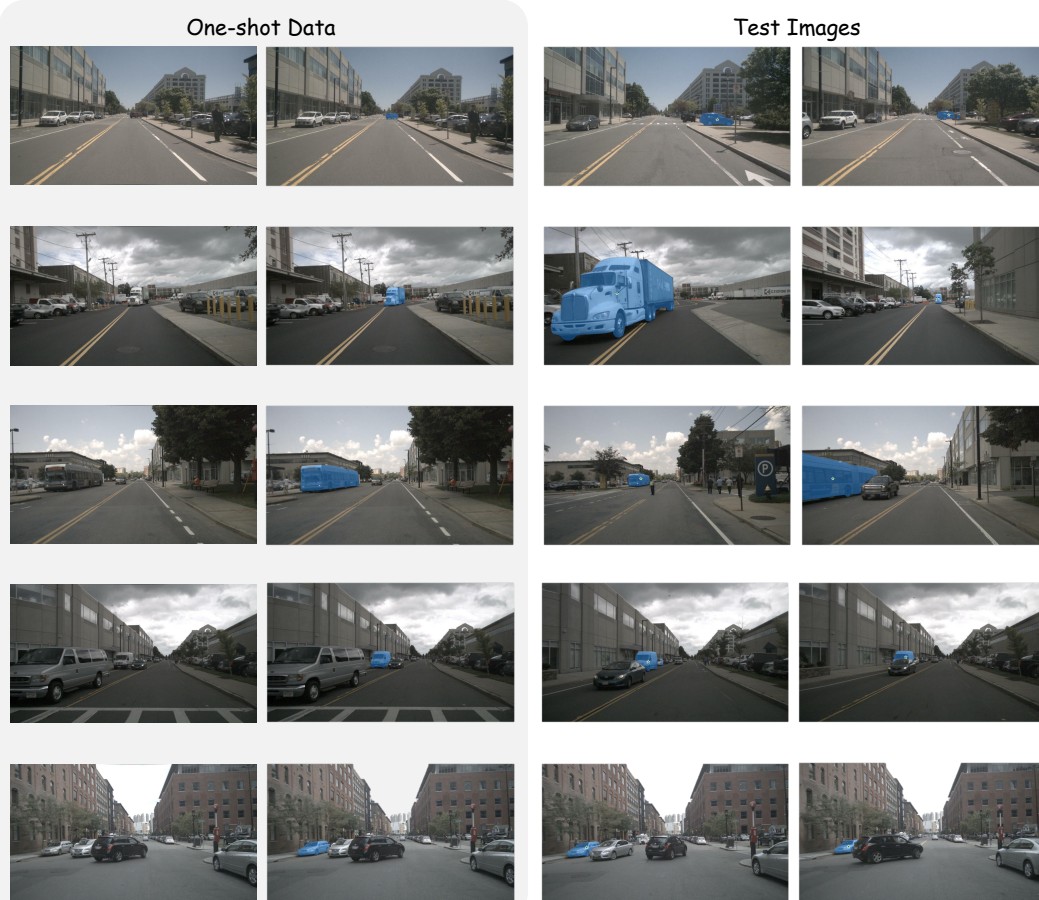

Figure 15: **One-shot segmentation of PerSAM-F in outdoor street scenes.**

As visualized in Figures 12, our method can precisely segment the dancing man in front of a crowd (the $2^{nd}$ row) and differentiate different fishes within a group (the last row).

### E.3    CAN PERSAM ALSO WORK ON SELF-DRIVING SCENARIOS?

***Yes.*** In most cases, our model can segment the designated cars with distinctive appearances in dense traffic. As visualized in Figure 15, for the user-provided target (e.g., a red car, a truck, and a bus), our PerSAM-F can well locate and segment them under severe occlusion or surrounded by similar cars.

### E.4    FAILURE CASES OF PERSAM-F

After solving the scale ambiguity issue, the three types of failure cases of PerSAM-F are shown in Figure 16: (a) different people with the same clothes, indicating our approach is not very sensitive to fine-grained human faces; (b) the key appearance of the target object is occluded by in test images (the red chest of the bird), indicating that we still need to improve our robustness when there is too large appearance change in test images; (c) discontinuous objects that SAM cannot tackle, for which we can replace SAM with stronger segmentation foundation model for assistance.

### E.5    CAN PERSAM SEGMENT MULTIPLE IDENTICAL OBJECTS IN AN IMAGE?

***Yes.*** As shown in Figure 17, given the one-shot image of a reference cat, if the test image contains two similar cats that are expected to be both segmented, we propose two simple strategies for PerSAM:

Failure Cases of PerSAM-F

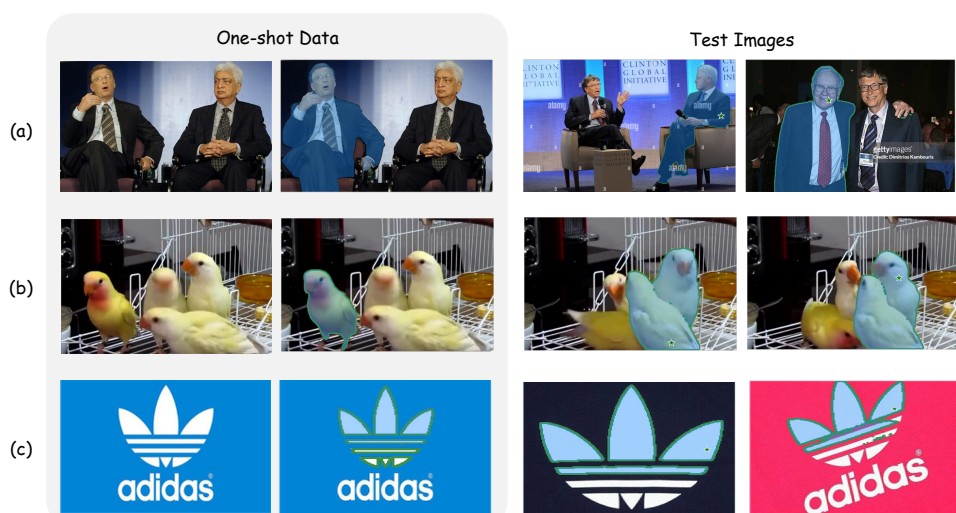

Figure 16: **Three types of failure cases of PerSAM-F.**

Table 18: **Statistic of Location Confidence Scores for Different Objects** in the PerSeg dataset.

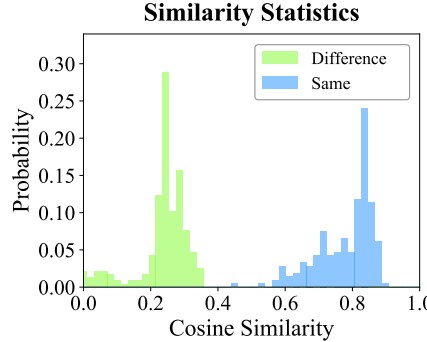

Table 19: **DreamBooth Assisted by PerSAM** with quantitative results. We adopt CLIP (Radford et al., 2021) to calculate the image-text and -image similarity.

| Method | Text-Align | Image-Align | KID ($\times 10^3$) |
|---|---|---|---|
| DreamBooth | 0.812 | 0.793 | 29.7 |
| + PerSAM | 0.830 | 0.814 | 29.2 |
| + PerSAM-F | **0.834** | **0.818** | **28.9** |

**Iterative Masking.** For two similar cats, we first calculate the location confidence map $S_1$, and utilize PerSAM to segment one of the cats, denoting the obtained mask prediction as $M_1$. Then, we reweigh the confidence map $S_1$ by assigning zeros to the area within $M_1$. We denote the masked confidence map as $S_2$. After this, we enable PerSAM's decoder to subsequently segment the second cat and acquire $M_2$. In this way, our approach can iteratively mask the already segmented objects and segment all the expected similar objects, until there is no target in the image.

**Confidence Thresholding.** How to stop the iteration when there is no other expected object in the image? We introduce a thresholding strategy for adaptive control. As shown by the statistics in Table 18, we count the confidence scores of the positive location prior (the maximum score on the confidence map) for two groups of objects in the PerSeg dataset: 'Same' and 'Different', where we utilize DINOv2 (Oquab et al., 2023) as the image encoder. 'Same' utilizes the same object for reference and test, just like the normal evaluation. 'Different' utilizes one object for reference, but tests on all other 39 objects. We observe the scores in 'Same' are almost all larger than 0.5, while those in 'Different' are lower than 0.4. Therefore, we adopt a simple thresholding strategy to stop the iterative segmentation based on the confidence map with a threshold of 0.45, which can well discriminate different objects or categories for most cases, e.g., segmenting all the cats or dogs in the image shown in Figure 18. In this way, for a test image, if the maximum score in the confidence map is lower than 0.45, there is no more target object in the image and we would stop the iteration.

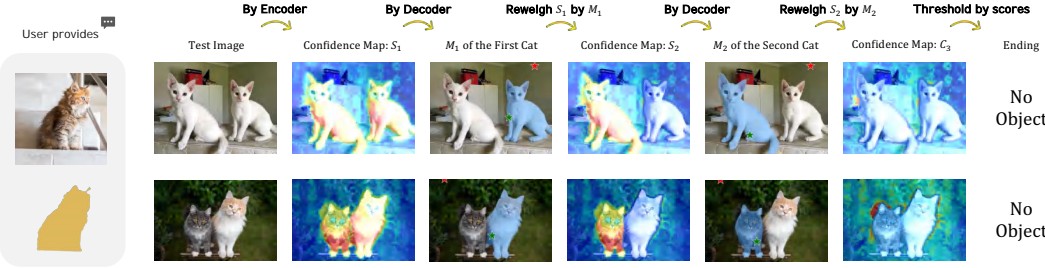

Figure 17: **Segmenting Multiple Similar Objects in an Image.** We adopt two strategies for PerSAM to simultaneously segment multiple similar objects: iterative masking and confidence thresholding. We denote the positive and negative location prior by green and red pentagrams, respectively.

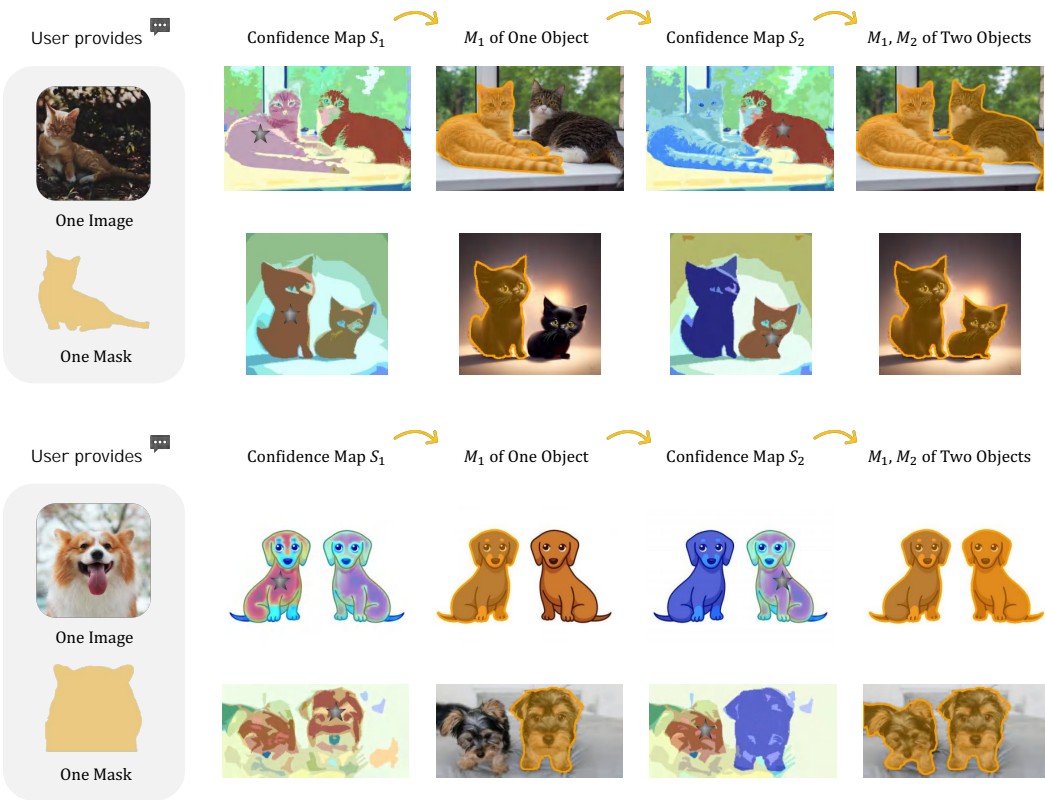

Figure 18: **Segmenting Objects of the Same Category.** Besides specific visual concepts, our approach can also be personalized by a category, cat or dog, with a confidence thresholding strategy.

### E.6 IS PERSAM-F GENERALIZED ONLY TO A SPECIFIC OBJECT?

Our PerSAM-F can not only be personalized by a specific object, but also generalize to a certain category with the same amount of parameters. As visualized in Figure 13, given a reference cone/armour/crocodile in FSS-1000 dataset (Li et al., 2020), our PerSAM-F can well segment other similar cones/armours/crocodiles in test images. This is because objects of the same category can contain similar hierarchical structures, so the learned scale weights of PerSAM-F by one sample can also be applicable to different objects within the same category. In contrast, for different categories, one needs to fine-tune two sets of scale weights to respectively fit their scale information.

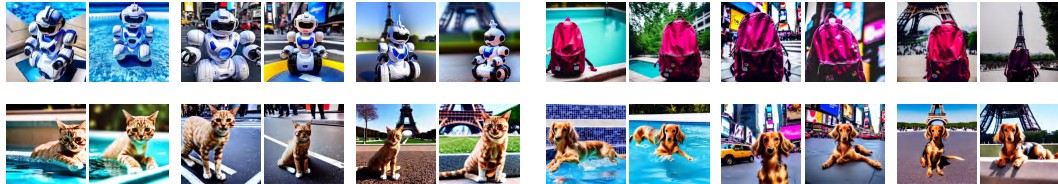

Figure 19: **Visualization of the Enlarged PerSeg Dataset** generated by a fine-tuned Dream-Booth (Ruiz et al., 2022). We show the examples of four objects with three different text prompts: 'A photo of an [OBJECT] in a swimming pool/in Times Square/in front of Eiffel Tower.'

Table 20: **Personalized Object Segmentation on the Enlarged PerSeg Dataset** with 5x largaer in size. We compare the overall mIoU and bIoU for different methods (Bar et al., 2022; Wang et al., 2022; 2023; Zou et al., 2023).

| Method | Painter | SEEM | SegGPT | PerSAM | PerSAM-F |
|---|---|---|---|---|---|
| mIoU | 43.6 | 82.8 | 87.8 | 85.9 | 89.6 |
| bIoU | 37.5 | 51.3 | 69.7 | 66.2 | 72.4 |

### E.7 WILL PERSAM BE CONSTRAINED BY SAM'S LIMITED SEMANTICS BY CLASS-AGNOSTIC TRAINING?

*Yes*, due to SAM's inherent class-agnostic training, the visual features extracted by SAM's encoder contain limited category-level semantics. This might constrain the category-level discriminative capability for complex multi-object scenes. Observing this limitation, we locate the target object among other objects in test images entirely by feature matching, i.e., the location confidence map. Such a matching strategy only considers the appearance-based class-agnostic similarity, without category semantics. To this end, we can leverage other semantically rich image encoders, e.g., CLIP (Radford et al., 2021) and DINOv2 (Oquab et al., 2023), for PerSAM(-F) to improve the multi-object performance. We conduct an ablation study of different image encoders on DAVIS 2017 dataset (Pont-Tuset et al., 2017) for video object segmentation, which contains multiple similar objects within a video. As shown in Table 15, applying CLIP and DINOv2 with more sufficient semantic knowledge can improve the results of PerSAM-F for more challenging multi-object segmentation.

### E.8 CAN PERSAM HELP DREAMBOOTH ACHIEVE BETTER MULTI-OBJECT CUSTOMIZATION?

*Yes.* Similar to single-object personalization, we only calculate the loss within foreground regions for DreamBooth (Ruiz et al., 2022) with multi-object training samples. As visualized in Figure 20, we show the improvement for two-object customization assisted by our PerSAM. The backgrounds within images generated by DreamBooth are severely disturbed by those within few-shot training images, while the PerSAM-assisted DreamBooth can accurately synthesize new backgrounds according to the input language prompts.

### E.9 SCALING PERSEG DATASET

Although our newly constructed PerSeg dataset contains different objects in various contexts, it is relatively small in scale compared to existing segmentation benchmarks. For a more robust evaluation, we enlarge the PerSeg dataset (40 objects with 5~7 images per object) to 30 images per object, **5x larger** in scale. We leverage the existing few-shot images to fine-tune DreamBooth (Ruiz et al., 2022) respectively for each object, and then generate new images with diverse backgrounds or poses (swimming pool, Times Square, Eiffel Tower, etc. . . . ), including richer data examples as shown in Figure 19. We report the segmentation results in Table 20, the scale-aware fine-tuned PerSAM-F still achieves the best performance, and the training-free PerSAM can also surpass Painter and SEEM, demonstrating the superior robustness of our approach.

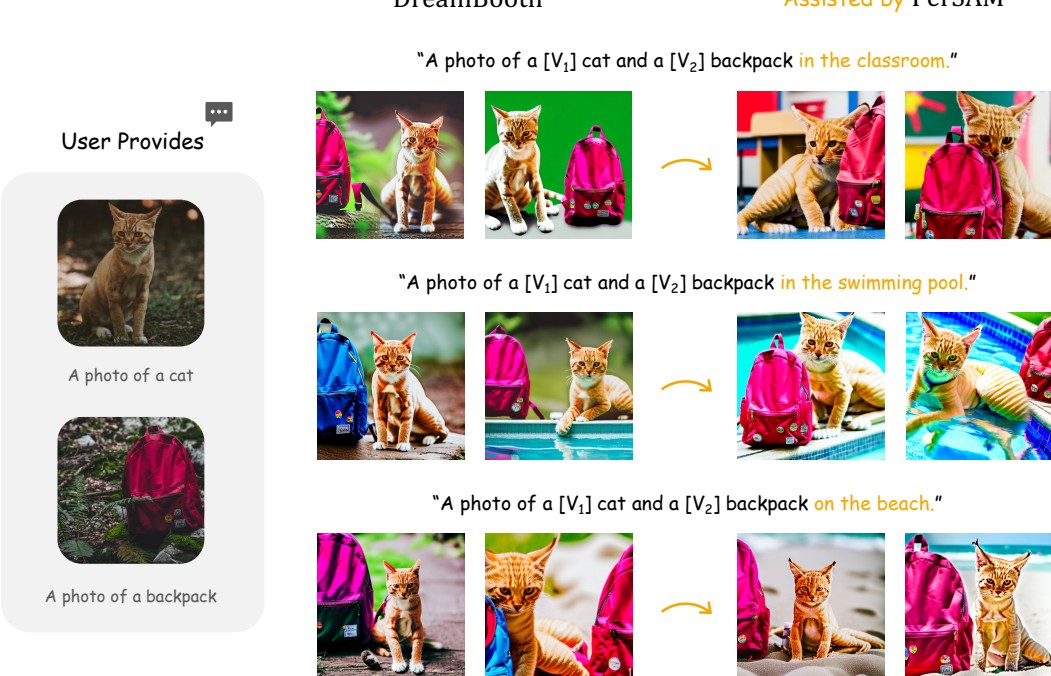

Figure 20: **Multi-object generation of PerSAM-assisted DreamBooth (Ruiz et al., 2022).**

## E.10 ANY OTHER APPLICATIONS FOR PERSAM?

For CLIP-based (Radford et al., 2021) few-shot image classification, a series of works (Zhang et al., 2021; 2023d; Udandarao et al., 2022) extract the visual features of few-shot images by CLIP, and cache them as category prototypes for downstream adaption of CLIP. However, such prototypes contain the visual noises of the backgrounds that disturb the category semantics. Therefore, our PerSAM is helpful in segmenting the objects of the same category in few-shot images, and enables the CLIP-based methods to cache only foreground informative features. For 3D reconstruction by NeRF (Mildenhall et al., 2021), existing approaches can only lift the objects, which are annotated with multi-view masks, into 3D space. Considering that the multi-view annotation is labor-intensive, our approach provides a solution for NeRF to lift any object in a scene, simply by prompting SAM to segment the object in one view. On top of that, PerSAM can be personalized to generate the masks in all multi-view images, allowing for efficient and flexible 3D reconstruction. We leave these applications as future works.

