# PERSONALIZE SEGMENT ANYTHING MODEL WITH ONE SHOT

## A  OVERVIEW

- Section B: Related work.
- Section C: Experimental details and visualization.
- Section D: Additional experiments and analysis.
- Section E: Additional discussion.

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

DreamBooth            Assisted by PerSAM

"A photo of a [V$_1$] cat and a [V$_2$] backpack in the classroom."

User Provides

A photo of a cat

A photo of a backpack

"A photo of a [V$_1$] cat and a [V$_2$] backpack in the swimming pool."

"A photo of a [V$_1$] cat and a [V$_2$] backpack on the beach."

Figure 10: **Multi-object text-to-image generation of PerSAM-assisted DreamBooth (Ruiz et al., 2022).**

Table 14: **Personalized Object Segmentation on the Enlarged PerSeg Dataset** with 5x largaer in size. We compare the overall mIoU and bIoU for different methods (Bar et al., 2022; Wang et al., 2022; 2023; Zou et al., 2023).

| Method | Painter | SEEM | SegGPT | PerSAM | PerSAM-F |
|--------|---------|------|--------|--------|----------|
| mIoU   | 43.6    | 82.8 | 87.8   | 85.9   | 89.6     |
| bIoU   | 37.5    | 51.3 | 69.7   | 66.2   | 72.4     |

dataset (Pont-Tuset et al., 2017) for video object segmentation, which contains multiple similar objects within a video. As shown in Table 9, applying CLIP and DINOv2 with more sufficient semantic knowledge can improve the results of PerSAM-F for more challenging multi-object segmentation.

### E.8   CAN PERSAM HELP DREAMBOOTH ACHIEVE BETTER MULTI-OBJECT CUSTOMIZATION?

***Yes.*** Similar to single-object personalization, we only calculate the loss within foreground regions for DreamBooth (Ruiz et al., 2022) with multi-object training samples. As visualized in Figure 10, we show the improvement for two-object customization assisted by our PerSAM. The backgrounds within images generated by DreamBooth are severely disturbed by those within few-shot training images, while the PerSAM-assisted DreamBooth can accurately synthesize new backgrounds according to the input language prompts.

### E.9   SCALING PERSEG DATASET

Although our newly constructed PerSeg dataset contains different objects in various contexts, it is relatively small in scale compared to existing segmentation benchmarks. For a more robust evaluation, we enlarge the PerSeg dataset (40 objects with 5∼7 images per object) to 30 images per object, **5x larger** in scale. We leverage the existing few-shot images to fine-tune DreamBooth (Ruiz et al.,

2022) respectively for each object, and then generate new images with diverse backgrounds or poses (swimming pool, Times Square, Eiffel Tower, etc. . . . ), including richer data examples as shown in Figure 9. We report the segmentation results in Table 14, the scale-aware fine-tuned PerSAM-F still achieves the best performance, and the training-free PerSAM can also surpass Painter and SEEM, demonstrating the superior robustness of our approach.

### E.10 ANY OTHER APPLICATIONS FOR PERSAM?

Besides improving the generation of DreamBooth (Ruiz et al., 2022), our PerSAM and PerSAM-F can also be utilized to assist other models and applications, such as CLIP (Radford et al., 2021) and NeRF (Mildenhall et al., 2021). For CLIP-based few-shot image classification, a series of works (Zhang et al., 2021; 2023d; Udandarao et al., 2022) extract the visual features of few-shot images by CLIP, and cache them as category prototypes for downstream adaption of CLIP. However, such prototypes contain the visual noises of the backgrounds that disturb the category semantics. Therefore, via the category-wise personalization approach, our PerSAM is helpful in segmenting the objects of the same category in few-shot images, and enables the CLIP-based methods to cache only foreground informative features. For 3D reconstruction by NeRF, existing approaches can only lift the objects, which are annotated with multi-view masks, into 3D space. Considering that the multi-view annotation is labor-intensive, our approach provides a solution for NeRF to lift any object in a scene, simply by prompting SAM to segment the object in one view. On top of that, PerSAM can be personalized to generate the masks in all multi-view images, allowing for efficient and flexible 3D reconstruction. We leave these applications as future works.