# OpenReview forum: "Personalize Segment Anything Model with One Shot"
_ICLR.cc/2024/Conference — ICLR 2024 poster_

### Official Review · Reviewer_inm4 · 2023-10-29

**Soundness:** 4 excellent
**Presentation:** 4 excellent
**Contribution:** 4 excellent
**Rating:** 8
**Confidence:** 5

**Summary:**

This paper proposes PerSAM, a novel training-free method, to customize the general-purpose SAM for personalized object segmentation by using a single image with a reference mask. Additionally, the paper introduces PerSAM-F, an efficient variant that enhances performance by tuning just two parameters within 10 seconds. The effectiveness of the proposed method is demonstrated by comprehensive experiments and ablation studies.

**Strengths:**

* This paper is well-written and easy to understand.

* This paper first studies an interesting task of customizing a general-purpose segmentation model for personalized scenarios. And the paper presents a highly effective method to address this task.

* The method is simple and easy to follow. The proposed PerSAM can guide SAM to segment target objects by three effective training-free techniques.  By tuning 2 parameters within 10 seconds, PerSAM-F efficiently alleviates the mask ambiguity issue and improves the performance.

* This paper has comprehensive discussions and experiments and shows good performances on various tasks.

**Weaknesses:**

The feature semantics of SAM might be limited due to SAM's class-agnostic training. While PerSAM and PerSAM-F demonstrate promising performance in personalized object segmentation, their effectiveness may be constrained by SAM's feature semantics in scenarios involving multiple different objects. This may require additional training to enable better transfer of SAM's features to downstream tasks. Alternatively, introducing other representations with stronger semantics, such as CLIP.

**Questions:**

* Can PerSAM be extended to do few-shot segmentation with more reference masks for achieving better performance? In the real scenario, utilizing 5-shot or 10-shot examples does not significantly increase the cost compared to 1-shot, but it does offer more precise visual information.

* Is the fine-tuned PerSAM-F generalized only to a specific object? Can PerSAM generalize to different objects using the same parameters? What about using different fine-tuning methods such as LoRA for multi-objects generalization performance?

* Can PerSAM help DreamBooth achieve more complex multi-object customization?

---

> ### Author Response · Authors · 2023-11-17
> **Response to Reviewer inm4**
>
> We sincerely appreciate your valuable reviews and recognition of our work. We hope our response can address your concerns.
>
> ---
> > **Q1: PerSAM and PerSAM-F may be constrained by SAM's limited semantics by class-agnostic training**
>
> >
> 1. Indeed, due to SAM's inherent class-agnostic training, the visual features extracted by SAM's encoder contain limited category-level semantics. This might constrain the category-level discriminative capability for complex multi-object scenes. Observing this limitation, we locate the target object among other objects in test images entirely by ***feature matching***, i.e., the location confidence map. Such a matching strategy only considers the appearance-based class-agnostic similarity, without category semantics.
>
> 2. As you suggested, we can leverage other semantically rich image encoders, e.g., CLIP and DINOv2, for PerSAM(-F) to improve the multi-object performance. We conduct an ablation study of different image encoders on DAVIS 2017 dataset for video object segmentation, which contains multiple similar objects within a video. As shown in the table below, applying CLIP and DINOv2 with more sufficient semantic knowledge can improve the results of PerSAM-F for more challenging multi-object segmentation.
> Encoder|J&F|J|F
> -|:-:|:-:|:-:
> SAM|73.4|70.9|75.9
> CLIP|74.9|72.8|77.0
> DINOv2|76.1|73.2|78.9
>
>
> ---
> > **Q2: Can PerSAM be extended to few-shot segmentation for better performance?**
>
> >
> Thanks for your advice! Yes, our approach is not limited to one-shot segmentation, and can accept few-shot references for improved results. As an example, given 3-shot references, we independently calculate 3 location confidence maps for the test image, and adopt a pixel-wise max pooling to obtain the overall location estimation. For PerSAM-F, we regard all 3-shot data as the training set to conduct the scale-aware fine-tuning.
>
> We respectively conduct experiments for 3-shot segmentation on ***PerSeg*** dataset and 5-shot segmentation on ***FSS-1000*** dataset. The results are shown in the following two tables. By providing more visual semantics in few-shot data, both our training-free PerSAM and the fine-tuned PerSAM-F can be further enhanced.
> Method|Shot|mIoU|bIoU
> -|:-:|:-:|:-:
> SegGPT|1-shot|94.3|76.5
> SegGPT|3-shot|96.7|78.4
> PerSAM|1-shot|89.3|71.7
> PerSAM|3-shot|90.2|73.6
> PerSAM-F|1-shot|95.3|77.9
> PerSAM-F|3-shot|97.4|79.1
>
> Method|Shot|mIoU
> -|:-:|:-:
> SegGPT|1-shot|85.6
> SegGPT|5-shot|89.3
> PerSAM|1-shot|81.6
> PerSAM|5-shot|82.3
> PerSAM-F|1-shot|86.3
> PerSAM-F|5-shot|89.8
>
> ---
> > **Q3: Is the fine-tuned PerSAM-F generalized only to a specific object?**
>
> >
> Our PerSAM-F can not only be personalized by a specific object, but also generalize to a certain category with the same amount of parameters. As visualized in ***Figure 13 of Appendix***, given a reference cone/armour/crocodile in FSS-1000 dataset, our PerSAM-F can well segment other similar cones/armours/crocodiles in test images. This is because objects of the same category can contain similar hierarchical structures, so the learned scale weights of PerSAM-F by one sample can also be applicable to different objects within the same category. In contrast, for different categories, one needs to fine-tune two sets of scale weights to respectively fit their scale information.
>
> ---
> > **Q4: What about using different fine-tuning methods, e.g., LoRA, for multi-object generalization?**
>
> >
> Thanks for your advice! As shown in ***Table 3 of the paper***, we have conducted an ablation study using different fine-tuning methods for single-object segmentation on PerSeg. In the table below, we further investigate different fine-tuning methods for multiple objects on DAVIS 2017 dataset. We respectively tune different parameters for different objects within a video. As shown, our proposed scale-aware fine-tuning performs the best with only 2 parameters.
> Method|Parameter|J&F
> -|:-:|:-:
> PerSAM|0|66.9
> Adapter|196K|50.4
> LoRA|293K|75.8
> PerSAM-F|2|76.1
>
> ---
> > **Q5: Can PerSAM help DreamBooth achieve better multi-object customization?**
>
> >
> Yes. Similar to single-object personalization, we only calculate the loss within foreground regions for DreamBooth with multi-object training samples. As visualized in ***Figure 20 of Appendix***, we show the improvement for two-object customization assisted by our PerSAM. The backgrounds within images generated by DreamBooth are severely disturbed by those within few-shot training images, while the PerSAM-assisted DreamBooth can accurately synthesize new backgrounds according to the input language prompts.

---

> > ### Comment · Reviewer_inm4 · 2023-11-21
> > **Thanks a lot for the clarification!**
> >
> > Thank you for your response, and pardon my late reply. I have read the rebuttal and other reviewers' comments and decided to keep my original score.

---

> > > ### Author Response · Authors · 2023-11-21
> > > **Thanks for your reply!**
> > >
> > > Thank you for acknowledging our response and efforts!

---

### Official Review · Reviewer_Fp1Y · 2023-10-30

**Soundness:** 3 good
**Presentation:** 3 good
**Contribution:** 2 fair
**Rating:** 6
**Confidence:** 5

**Summary:**

The paper proposes a training-free Personalisation approach for SAM. For PerSAM, it uses a reference mask for obtaining the positive-negative location prior and then enable SAM for segmentation by: 1) target-guided attention and 2) target-semantic prompting. To reduce the ambiguity of segmentation scales, PerSAM-F is further proposed to fine-tuning SAM. PerSeg dataset is also constructed to evaluate the personalisation segmentation performance. The paper is also evaluated on the one-shot image and video segmentation benchmarks.

**Strengths:**

1. The paper is well organised with clear motivation and easy to understand. The illustration and visualisation figures are well presented.

2. PerSAM is training-free and computationally efficient, where the ablation experiment for PerSAM in Table 4, 5 and 6 are extensive.

3. The paper demonstrates good performance not only on the constructed PerSeg benchmark, but also on many image/video segmentation benchmarks.

**Weaknesses:**

1. In the appendix, the author mentioned using dinov2 features. Can the authors also provide the results in Table 2 and 3 by using the default image encoder features of SAM?

2. What is the running speed/ memory consumption of PerSAM comparing to SAM?

3. In Table 2, can the author provide performance comparison to SAM-PT [a]? [a] is a related work in adapting SAM for video object segmentation.

[a] SAM-PT: Extending SAM to zero-shot video segmentation with point-based tracking. arXiv, 2023.

4. On the video object segmentation benchmark, how dose PerSAM differentiate objects with similar appearances? Can PerSAM also work on self-driving scenario with many similar vehicles/cars in the dense traffic?

**Questions:**

Can the paper provide more failure cases visualisation for PerSAM? I am generally positive about this paper. If my concerns in the weakness section can be addressed, I will consider to further upgrade my rating.

---

> ### Author Response · Authors · 2023-11-17
> **Response to Reviewer Fp1Y (Part #1)**
>
> We sincerely appreciate your careful and constructive reviews. We hope our response can address your concerns.
>
> ---
> > **Q1: What are the results in Tables 2 and 3 by using the default image encoder features of SAM?**
>
> >
> Thanks for your advice! The detailed results of the two image encoders in Tables 2 and 3 are respectively as follows.
> Method|Encoder|J&F|J|F
> -|:-:|-|-|-
> Painter|-|34.6|28.5|40.8
> SEEM|-|58.9|55.0|62.8
> SegGPT|-|75.6|72.5|78.6
> PerSAM|SAM|62.8|58.6|67.0
> PerSAM|DINOv2|66.9|63.4|70.4
> PerSAM-F|SAM|73.4|70.9|75.9
> PerSAM-F|DINOv2|76.1|73.2|78.9
>
>
>
> Method|Encoder|FSS-1000|LVIS-92$^i$|PASCAL-Part|PACO-Part
> -|:-:|:-:|:-:|:-:|:-:
> Painter|-|61.7|10.5|30.4|14.1
> SegGPT|-|85.6|18.6|-|-
> PerSAM|SAM|74.9|12.9|31.3|21.2
> PerSAM|DINOv2|81.6|15.6|32.5|22.5
> PerSAM-F|SAM|79.4|16.2|32.0|21.3
> PerSAM-F|DINOv2|86.3|18.4|32.9|22.7
>
> As DINOv2 is particularly pre-trained by large-scale contrastive data, it produces more discriminative image features than SAM's encoder. This contributes to a more precise positive-negative location prior for better segmentation results, especially on the challenging FSS-1000 dataset. Despite this, with SAM's original encoder, our PerSAM-F and the training-free PerSAM still obtain better segmentation accuracy than Painter or SEEM on different datasets, demonstrating the effectiveness of our approach.
>
> ---
> > **Q2: What's the running speed/memory compared to SAM?**
>
> >
>
> We test the running efficiency on a single NVIDIA A100 GPU with batch size 1. As shown in the table below, our PerSAM and PerSAM-F bring marginal latency and GPU memory consumption over SAM.
> Method|FPS$\uparrow$|Memory (MB)$\downarrow$
> -|:-:|:-:
> SAM|2.16|5731
> PerSAM|2.08|5788
> PerSAM-F|1.98|5832
>
> ---
> > **Q3: How's the performance comparison to SAM-PT?**
>
> >
>
> Thanks for pointing out! We will add the discussion and comparison in the revised version.
>
> 1. ***Method Comparison.*** Although both our PerSAM(-F) and SAM-PT are developed based on SAM, our approach can be generalized to most one-shot segmentation tasks (personalized/video/semantic/part segmentation), while SAM-PT specifically aims at video object segmentation. One key difference between our approach and SAM-PT is how to locate and associate objects from the previous to the current frame, i.e., propagating the location prompt for SAM across frames. In detail, our PerSAM(-F) simply calculates a location confidence map by feature matching, while SAM-PT relies on an external point tracking network, PIPS [1].
>
> 2. ***Performance Comparison.*** As shown in the table below, on DAVIS 2017 dataset, SAM-PT performs slightly better than the original PerSAM-F. However, inspired by SAM-PT, we can also incorporate its point tracking strategy (the PIPS tracker) with PerSAM(-F)  to propagate the positive-negative point prompt, which effectively enhances the segmentation performance. This demonstrates the flexible extensibility of our approach for applying more advanced trackers in a plug-and-play way.
> Method|Propagation|J&F
> -|-|:-:
> SAM-PT|Point Tracking|76.6
> PerSAM|Feature Matching|66.9
> PerSAM|+Point Tracking|68.2
> PerSAM-F|Feature Matching|76.1
> PerSAM-F|+Point Tracking|77.2
>
> #### Reference:
> #### [1] Particle Video Revisited: Tracking through Occlusions using Point Trajectories. ECCV 2022.

---

> ### Author Response · Authors · 2023-11-17
> **Response to Reviewer Fp1Y (Part #2)**
>
> > **Q4: How does PerSAM differentiate objects with similar appearances in video object segmentation?**
>
> >
>
> For video object segmentation, our approach tries to accurately locate the target object among similar ones by the following three aspects.
>
> 1. ***Discriminative features from the encoder.*** Due to large-scale pre-training, the SAM's image encoder, or the more powerful DINOv2, can already produce discriminative enough visual features for different similar objects, which is fundamental to the subsequent calculation of location confidence map.
>
> 2. ***Comprehensive location confidence map.*** We calculate a set of confidence maps for all foreground pixels within the target object, such as the head, the body, or the paws of a dog, and then aggregate them to obtain an overall location estimation. This strategy can comprehensively consider the slight differences in any local parts between similar objects.
>
> 3. ***Temporal cues between adjacent frames.*** To better leverage the temporal consistency along the video, we prompt SAM's decoder additionally with the object bounding box from the last frame. As different objects have different trajectories, such temporal constraints can better differentiate similar objects by spatial locations.
>
> As visualized in ***Figure 12 of Appendix***, our method can precisely segment the dancing man in front of a crowd (the 2$^{nd}$ row) and differentiate different fishes within a group (the last row).
>
> ---
> > **Q5: Can PerSAM also work on self-driving scenarios with many similar vehicles/cars?**
>
> >
>
> Yes. In most cases, our model can segment the designated cars with distinctive appearances in dense traffic.
>
> 1. As visualized in ***Figure 15 of Appendix***, for the user-provided target (e.g., a red car, a truck, and a bus), our PerSAM-F can well locate and segment them under severe occlusion or surrounded by similar cars.
>
> 2. We conduct an experiment for one-shot segmentation on ***Tokyo Multi-Spectral-4$^i$*** [2] dataset, which contains 16 classes within outdoor self-driving scenarios. As shown in the table below, our PerSAM-F can surpass existing methods that require in-domain training, indicating our generalization capacity in city traffic scenes.
> Method|In-domain Train|mIoU
> -|:-:|:-:
> PFENet|$\checkmark$|14.0
> PGNet|$\checkmark$|17.5
> V-TFSS|$\checkmark$|26.1
> PerSAM|-|18.4
> PerSAM-F|-|25.6
>
> ---
> > **Q6: More failure cases visualization of PerSAM**
>
> >
>
> Thanks for your advice! There are mainly two kinds of failure cases of PerSAM(-F).
>
> 1. As shown in ***Figures 2 and 8 of the paper***, the training-free PerSAM might fail to segment objects with hierarchical structures, e.g., the hat on top of a teddy bear, or the head of a robot toy. Such scale ambiguity can be effectively alleviated by the fine-tuning of PerSAM-F.
>
> 2. After solving the scale ambiguity issue, the three types of failure cases of PerSAM-F are shown in ***Figure 16 of Appendix***: (a) different people with the same clothes, indicating our approach is not very sensitive to fine-grained human faces; (b) the key appearance of the target object is occluded by in test images (the red chest of the bird), indicating that we still need to improve our robustness when there is too large appearance change in test images; (c) discontinuous objects that SAM cannot tackle, for which we can replace SAM with stronger segmentation foundation model for assistance.
>
> #### Reference:
> #### [2] Visible and Thermal Images Fusion Architecture for Few-shot Semantic Segmentation. JVCIR 2021.

---

> > ### Comment · Reviewer_Fp1Y · 2023-11-21
> > **Thanks for the rebuttal clarification**
> >
> > Most of my concerns have been well addressed after the rebuttal. I also read the comments by other reviewers. Thus, I keep my rating.

---

> > > ### Author Response · Authors · 2023-11-21
> > > **Thanks for you reply!**
> > >
> > > Thank you for acknowledging our response and efforts!

---

### Official Review · Reviewer_XA1U · 2023-11-01

**Soundness:** 3 good
**Presentation:** 3 good
**Contribution:** 3 good
**Rating:** 6
**Confidence:** 4

**Summary:**

The paper presents a lightweight method to leverage the Segment Anything Model (SAM), to perform single shot image segmentation. SAM is a powerful image segmentation framework, however it is becomes challenging to use it to perform personalized image segmentation by manually tuning its input prompts. In the approach proposed in the paper, the authors present Personalized approach for SAM (PerSeg) which is training free and can be used to segment a particular object (class) across images given only a single example image along with its binary segmentation mask. To do this, the authors present two approaches: (1) target-guided attention, and (ii) target-semantic prompting. Additionally, to alleviate the problem of objects being present in different scales the paper presents a simple fine-tuning technique that keeps SAM frozen, to combine information from multi-scale masks and improve the final segmentation outputs.

A new dataset PerSeg is also introduced to test the proposed method, and finally the paper also shows how PerSAM can be used to improve DreamBooth [1], for the task of custom text-to-image synthesis.

The paper quantitatively and qualitatively demonstrates the efficacy of the proposed method across multiple datasets, and achieves competitive performance as compared to existing methods.

[1] Nataniel Ruiz, Yuanzhen Li, Varun Jampani, Yael Pritch, Michael Rubinstein, and Kfir Aberman. Dreambooth: Fine tuning text-to-image diffusion models for subject-driven generation. arXiv preprint arXiv:2208.12242, 2022

**Strengths:**

The problem of single-shot image segmentation is an important problem to solve. This has many downstream utilities in real-world applications ranging from design to healthcare. And the paper introduces a simple but effective technique to solve this by leveraging the powerful Segment Anything Module (SAM) [1].

The introduced method is called Personalization approach for SAM (PerSAM), and it takes as input a single example image of the desired object we want to segment, and its corresponding segmentation mask. This is then used to segment out the given object across multiple images automatically. To do this PerSeg involves 2 approachs:
(1) target-guided attention
(2) target semantic prompting

Additionally to handle the object occurring in different scales, the paper introduces a lightweight fine-tuning technique that keeps the SAM model frozen, and aggregates information across multiple scales for finer segmentation.

A new testing dataset is also introduced called PerSeg.

The work also shows the utility of the proposed method for improving text-to-image synthesis [2].

And finally the authors show qualitative and quantitative results for the proposed method, and it performs competitively as compared to existing approaches.

Overall a nicely written paper, with a simple idea and good results.


[1] Alexander Kirillov, Eric Mintun, Nikhila Ravi, Hanzi Mao, Chloe Rolland, Laura Gustafson, Tete Xiao, Spencer Whitehead, Alexander C Berg, Wan-Yen Lo, et al. Segment anything. arXiv preprint arXiv:2304.02643, 2023
[2] Nataniel Ruiz, Yuanzhen Li, Varun Jampani, Yael Pritch, Michael Rubinstein, and Kfir Aberman. Dreambooth: Fine tuning text-to-image diffusion models for subject-driven generation. arXiv preprint arXiv:2208.12242, 2022

**Weaknesses:**

Overall it is a nicely written paper, with good results.

However, it is somewhat lacking in it's quantitative evaluation. The choice of evaluation datasets is limited.

It would be worthwhile to also see the performance of the proposed method for one-shot segmentation on additional (more challenging) datasets like- MS-COCO, AED20K, CityScapes to also compare with more powerful existing state of the art models.

Also the comparison is lacking. It would be nice to compare against methods that do zero-shot or text guided segmentation like DenseCLIP [1].

[1]. DenseCLIP: Language-Guided Dense Prediction with Context-Aware Prompting, Yongming Rao, Wenliang Zhao, Guangyi Chen, Yansong Tang, Zheng Zhu, Guan Huang, Jie Zhou, Jiwen Lu, https://arxiv.org/abs/2112.01518

**Questions:**

How comparable is this work or results against methods that do zero-shot or text guided segmentation like DenseCLIP [1]? And why / why not does it make sense to compare against such methods?

[1]. DenseCLIP: Language-Guided Dense Prediction with Context-Aware Prompting, Yongming Rao, Wenliang Zhao, Guangyi Chen, Yansong Tang, Zheng Zhu, Guan Huang, Jie Zhou, Jiwen Lu, https://arxiv.org/abs/2112.01518

---

> ### Author Response · Authors · 2023-11-17
> **Response to Reviewer XA1U**
>
> We sincerely appreciate your detailed and insightful reviews. We hope our response can address your concerns.
>
> ---
> > **Q1: Experiments for one-shot segmentation on additional (more challenging) datasets**
>
> >
> Thanks for your advice! It is very necessary to evaluate PerSAM in more challenging scenarios.
>
> In our paper, we have experimented on ***six datasets*** of different tasks, i.e., ***PerSeg*** (personalized object segmentation), ***DAVIS 2017*** (video object segmentation), ***FSS-1000*** and ***LVIS-92$^i$*** (semantic segmentation), ***PASCAL-Part*** and ***PACO-Part*** (part segmentation). On different benchmarks with various domains, our PerSAM(-F) without in-domain training can attain competitive performance to existing specialist models.
>
> As you suggested, we further present the performance of our approach on two additional datasets for one-shot segmentation: ***COCO-20$^i$*** [1] and ***Tokyo Multi-Spectral-4$^i$*** [2]. (Note that, ADE20K and CityScapes have no corresponding few-shot benchmarks.)
>
> 1. ***COCO-20$^i$*** is constructed from MS-COCO by dividing the diverse 80 classes evenly into 4 folds. We directly test our method on the validation set without specific in-domain training. As shown in the below table, our PerSAM(-F) achieves favorable segmentation performance over a wide range of object categories, comparable to previous in-domain methods.
> Method|In-domain Train|mIoU
> -|:-:|:-:
> FPTrans|$\checkmark$|47.0
> SCCAN|$\checkmark$|48.2
> HDMNet|$\checkmark$|50.0
> PerSAM|-|47.9
> PerSAM-F|-|50.6
>
> 2. ***Tokyo Multi-Spectral-4$^i$*** is sampled from Tokyo Multi-Spectral containing 16 classes within outdoor city scenes, similar to CityScapes. Different from existing methods, we only take as input the RGB images without the paired thermal data, and do not conduct in-domain training. As shown in the table below, our approach still exhibits good generalization capacity in street scenarios.
> Method|In-domain Train|mIoU
> -|:-:|:-:
> PFENet|$\checkmark$|14.0
> PGNet|$\checkmark$|17.5
> V-TFSS|$\checkmark$|26.1
> PerSAM|-|18.4
> PerSAM-F|-|25.6
>
> ---
> > **Q2: Comparison with zero-shot or text-guided segmentation methods, like DenseCLIP**
>
> >
>
> Thanks for your advice! It would be beneficial to discuss the relationship between PerSAM and existing zero-shot text-guided segmentation models. However, please note that DenseCLIP cannot perform zero-shot text-guided segmentation for comparison with our method. We first explain the reason, and then show the comparison with other zero-shot text-guided methods.
>
> 1. ***Why DenseCLIP is not comparable?*** This is because DenseCLIP is a conventional closed-set method, which *cannot conduct zero-shot or text-guided segmentation*. Specifically, DenseCLIP only utilizes CLIP's text embeddings as axillary semantics for image features, and still requires fully supervised training for segmentation on limited "seen" classes. Therefore, it cannot be directly utilized for text-guided zero-shot segmentation on "unseen" classes, or one-shot segmentation given reference data.
>
> 2. ***Comparing with OVSeg [3] and Grounded-SAM [4].*** We select two popular text-guided open-set segmentation methods for comparison with our PerSAM(-F) on three benchmarks: ***PerSeg*** and ***COCO-20$^i$***. ***OVSeg*** leverages MaskFormer to first generate class-agnostic mask proposals, and then adopts a fine-tuned CLIP for zero-shot classification. ***Grounded-SAM*** utilizes a powerful text-guided detector, Grounding DINO [5], to generate object bounding boxes, and then utilize them to prompt SAM for segmentation. Instead of giving a one-shot reference, we directly prompt them by the category name of the target object for text-guided segmentation, e.g., "cat", "dog", or "chair". As shown in the table below, our PerSAM-F consistently achieves competitive results in the two scenarios. This indicates that utilizing PerSAM with a class-agnostic one-shot reference is on par with recognizing the category and then segmenting it with text-guided methods.
> Method|Prompt|PerSeg|COCO-20$^i$
> -|:-:|:-:|:-:
> OVSeg|Category Name|76.5|37.8
> Grounded-SAM|Category Name|93.2|49.9
> PerSAM|One-shot Data|89.3|47.9
> PerSAM-F|One-shot Data|95.3|50.6
>
>
> #### Reference:
> #### [1] Feature Weighting and Boosting for Few-shot Segmentation. ICCV 2019.
> #### [2] Visible and Thermal Images Fusion Architecture for Few-shot Semantic Segmentation. JVCIR 2021.
> #### [3] Open-Vocabulary Semantic Segmentation With Mask-Adapted CLIP. CVPR 2023.
> #### [4] https://github.com/IDEA-Research/Grounded-Segment-Anything.
> #### [5] Grounding DINO: Marrying DINO with Grounded Pre-Training for Open-Set Object Detection. arXiv 2023.

---

> > ### Comment · Reviewer_XA1U · 2023-11-21
> > **Response to authors.**
> >
> > Thanks a lot for the detailed response. I have analyzed the review in light of the response, and decide to keep my score.

---

> > > ### Author Response · Authors · 2023-11-21
> > > **Thanks for your reply!**
> > >
> > > Thank you for acknowledging our rebuttal and efforts!

---

### Meta-Review · Area_Chair_Dbim · 2023-12-02

**Metareview:**

The authors propose an efficient one-shot fine-tuning SAM for personalized segment anything (PerSAM-F). The finetuning only involves 2 parameters and takes 10 seconds to improve performance. PerSAM-F achieves the best performance in an in-house dataset, Video Object Segmentation (DAVIS 2017), several One-shot Semantic and Part Segmentation datasets, and an additional dataset for one-shot segmentation (COCO-20).
Pros:
* Based on SAM, the proposed method is simple and efficient.
* Performance has been validated on many datasets.
Cons:
* The performance improvement is not significant.
* limited novelty

Many additional experiments are conducted as requested by reviewers.
Hence, reviewers are happy to keep their positive ratings.
AC also recommends acceptance.

**Justification For Why Not Higher Score:**

The performance improvement is not significant.
On LVIS-92, it is even slightly worse than SegGPT.
Compared to SAM-PT, point tracking seems to be more important than the proposed components.

**Justification For Why Not Lower Score:**

Although the novelty is limited, performance of the proposed method has been validated on many datasets.
The reviewers also did not find big concerns of the paper.

---

### Decision · Program_Chairs · 2024-01-16

Accept (poster)